



# 1 Input-adaptive linear mixed-effects model for estimating alveolar
# 2 Lung Deposited Surface Area (LDSA) using multipollutant datasets

Pak Lun Fung[1,2], Martha A. Zaidan[1,3], Jarkko V. Niemi[4], Erkka Saukko[5], Hilkka Timonen[6], Anu Kousa[4],
Joel Kuula[6], Topi Rönkkö[7], Ari Karppinen[6], Sasu Tarkoma[8], Markku Kulmala[1,3], Tuukka Petäjä[1,3] and
Tareq Hussein[1,9]
[1]Institute for Atmospheric and Earth System Research / Physics, Faculty of Science, University of Helsinki, Finland
[2]Helsinki Institute of Sustainability Science, Faculty of Science, University of Helsinki, Finland
[3]Joint International Research Laboratory of Atmospheric and Earth System Sciences, School of Atmospheric Sciences, Nanjing
University, Nanjing 210023, China
[4]Helsinki Region Environmental Services Authority (HSY), P.O. Box 100, FI-00066 Helsinki, Finland
[5]Pegasor Ltd., FI-33100 Tampere, Finland
[6]Atmospheric Composition Research, Finnish Meteorological Institute, FI-00560 Helsinki, Finland
[7]Aerosol Physics Laboratory, Physics Unit, Tampere University, FI-33720 Tampere, Finland
[8]Department of Computer Science, Faculty of Science, University of Helsinki, Finland
[9]Department of Physics, the University of Jordan, Amman 11942, Jordan
*Correspondence to*: Pak Lun Fung (pak.fung@helsinki.fi), Tareq Hussein (tareq.hussein@helsinki.fi)
**Abstract.** Lung deposited surface area (LDSA) has been considered to be a better metric to explain nanoparticle toxicity
instead of the commonly used particulate mass concentration. LDSA concentrations can be obtained either by direct
measurements or by calculation based on the empirical lung deposition model and measurements of particle size distribution.
However, the LDSA or size distribution measurements are neither compulsory nor regulated by the government. As a result,
LDSA data are often scarce spatially and temporally. In light of this, we develop a novel statistical model, named input-
adaptive mixed-effects (IAME) model, to estimate LDSA based on other already existing measurements of air pollutant
variables and meteorological conditions. During the measurement period in 2017–2018, we retrieved LDSA data measured by
Pegasor AQ Urban and other variables at a street canyon (SC, average LDSA = 19.7±11.3 µm² cm⁻³) site and an urban
background (UB, average LDSA = 11.2±7.1 µm² cm⁻³) site in Helsinki, Finland. For the continuous estimation of LDSA,
IAME model is automatised to select the best combination of input variables, including a maximum of three fixed effect
variables and three time indictors as random effect variables. Altogether, 696 sub-models were generated and ranked by the
coefficient of determination ($R^2$), mean absolute error ($MAE$) and centred root-mean-square differences ($cRMSD$) in order. At
the SC site, the LDSA concentrations were best estimated by mass concentration of particle of diameters smaller than 2.5 µm
($PM_{2.5}$), total particle number concentration (PNC) and black carbon (BC), all of which are closely connected with the vehicular
emissions. At the UB site the LDSA concentrations were found to be correlated with $PM_{2.5}$, BC and carbon monoxide (CO).
The accuracy of the overall model was better at the SC site ($R^2$ = 0.80, $MAE$ = 3.7 µm² cm⁻³) than at the UB site ($R^2$ = 0.77,
$MAE$ = 2.3 µm² cm⁻³) plausibly because the LDSA source was more tightly controlled by the close-by vehicular emission
source. The results also demonstrate that the additional adjustment by taking random effects into account improves the
sensitivity and the accuracy of the fixed effect model. Due to its adaptive input selection and inclusion of random effects,
IAME could fill up missing data or even serve as a network of virtual sensors to complement the measurements at reference
stations.

## 38 1 Introduction

Particulate matter is one of the key components determining urban air pollution. Particulate matter can be described by a
combination of varying concentration (number, surface area and mass) and chemical composition. The mass concentrations of
particulate matter are dominated by large particles whereas the number concentrations are governed by sub-micron particles
(particle diameter ($d_p$) <1 µm), particularly ultrafine particles (UFP, $d_p$< 0.1 µm) (e.g. Petäjä et al., 2007; Rönkkö et al., 2017;



Zhou et al., 2020). Particulate matter of varying sizes, carrying various harmful substances, have been known for playing a
major role in adverse health effects (Dockery et al., 1993; Oberdorster, 2012; Shiraiwa et al., 2017) in particular to respiratory
system. A particle could be deposited in lung airways upon inhalation (Oberdörster et al., 2005) through three main
mechanisms: inertial impaction, gravitational sedimentation and Brownian diffusion. Interception, and electrostatic forces are
to a lesser extent. An airborne particle might be inhaled either through nasal or oral passage and enter the respiratory tract.
Coarser particles (5–30 μm) are usually partly deposited in the head airway by the inertial impaction mechanism because they
cannot follow the air streamline. Some finer particles (1–5 μm) are deposited in the tracheobronchial region, mainly through
gravitational sedimentation while some are removed by mucociliary clearance (Gupta and Xie, 2018). The remaining sub-
micron particles diffuse by Brownian motion and penetrate deeply into the alveolar region, which is considered to be the most
vulnerable section in lungs because removal mechanisms might be insufficient (Gupta and Xie, 2018). Inhaled particulate
matter could also function as a carrier, or as a transport vector, for many viruses, including the SARS-CoV-2 virus (COVID-
19, Prather et al., 2020), which is responsible for the pandemic recently declared by the World Health Organization (WHO).
Particulate matter may, therefore, increase the effectiveness of the virus spread in the aerosol as it creates a microenvironment
suitable for its persistence (Liu et al., 2018a) . Regular exposure to particulate matter increases the chance to suffer from acute
and chronic diseases (Brown et al., 2001; Oberdörster et al., 2005), and the susceptibility and severity of the COVID-19
patients' symptoms (Fennelly, 2020). In light of this, besides commonly monitored particulate matter number concentration
and mass concentration, the surface area of a particle is also an important factor when considering the harmfulness of
particulate matter (Duffin et al., 2002). In particular, the total surface area of particles which are deposited in alveolar section
of human lungs, known as Lung Deposited Surface Area (LDSA), is of the greatest concern because in vitro nanoparticle
toxicity has been demonstrated to be better explained when the lung burden was expressed as total particle surface area instead
of atmospheric particulate matter mass (e.g. Brown et al., 2001; Oberdorster, 2012; Schmid and Stoeger, 2016).

LDSA can be considered as an intermediary parameter between particle mass and particle number concentration as it cannot
be simply inferred from either of those parameters. Moreover, due to the various deposition efficiency with respect to particle
sizes, the quantification of LDSA is not simple. Conventionally, LDSA concentrations can be retrieved by (1) derivation from
particle size distribution with a deposition model or (2) direct measurements.

By fitting experimental lung deposition data on human beings, empirical deposition models are developed with the use of the
lung deposition model modified by Yeh and Schum (1980). Examples include the International Commission on Radiological
Protection (ICRP) Human Respiratory Tract Model (ICRP, 1994), the NCRP model (NCRP, 1997) and Multiple Path Particle
Dosimetry (MPPD) model (Anjilvel and Asgharian, 1995). Different conceptual particle deposition models vary primarily
with respect to lung morphometry and mathematical modelling techniques, rather than by using different deposition equations.
The three whole lung deposition models define regions of the human lungs (head airway, tracheobronchial and alveolar) for
any combination of particle size and breathing pattern (Hofmann, 2009). Among all models, single-path models, such as ICRP
model, are often used over multiple-path models due to their simplicity and their applicability to an average path without
requiring detailed knowledge of the branching structure of lungs. Owing to a higher potential health risk, LDSA in alveolar
region is often of greatest concern and it can be calculated by summing up the products of the surface concentration across
particle size spectrum and their corresponding deposition efficiency based on the selected deposition model.

Apart from numerical computation method, LDSA could also be measured by accredited instruments. LDSA concentration in
many urban environments is mainly driven by the particles smaller than 400 nm (Asbach et al., 2009; Kuuluvainen et al.,
2016), generated vastly by anthropogenic sources such as vehicular exhaust emissions (Karjalainen et al., 2016) and residential
wood combustion (Tissari, 2008) which typically produce large amount of small particles. The impact of larger particles (>400



nm) might be significant due to regional background in very polluted cities (e.g. Delhi, Salo et al., 2021a) or very low-quality
residential burning in detached housing areas (e.g. HMA, Pirjola et al., 2017). These small particles cannot be measured
precisely with methods relying solely on optical detection (e.g. no artificial growing of particles) as the light scattering intensity
of these particles is weak (Kulkarni et al., 2011). Hence alternative approaches are required. One approach is filter sampling
of aerosolised material followed by gas adsorption method (e.g. Lebouf et al., 2011). Another more common approach is using
a diffusion charging based technique where particles are charged with a unipolar corona charger (Fissan et al., 2006). This
method enables measurement of ultrafine particles and, more specifically, the LDSA concentration with good accuracy (Todea
et al., 2015) and stable performance in long term measurements (Rostedt et al., 2014). Nanoparticle Surface Area Monitor
(NSAM) has been used for decades (e.g. Asbach et al., 2009; Hama et al., 2017; Kiriya et al., 2017; Hennig et al., 2018), and
several other instruments and sensors, including DiSCmini, Testo Inc. (e.g. Eeftens et al., 2016; Habre et al., 2018) and
Partector, Naneos Ltd. (e.g. Cheristanidis et al., 2020), and Pegasor AQ Urban, Pegasor Ltd. (e.g. Kuuluvainen et al., 2018;
Kuula et al., 2020), using similar measuring techniques, are developed later on. Recently, this diffusion charging based LDSA
measurement has been combined with electrical cascade impactor method, which enables high time resolution measurements
of particle LDSA size distributions (Lepistö et al., 2020). Using these instruments in campaigns and continuous measurements,
LDSA concentrations and size distribution measurements in various environments have been reported across the globe in the
past decade (**Table 1**). When comparing LDSA concentrations measured by different instruments, it should be noted that the
instruments' limitation should be taken into account in experimental LDSA studies, which will be further discussion in Sect.
103   2.2.
Although each of these methods is capable of measuring aerosol surface area concentrations, the corresponding uncertainties
(Asbach et al., 2017) and cost hinder the widespread use in monitoring networks. Even though the instruments are available,
missing data often takes place due to instruments maintenance and data corruption. Kuula et al. (2020) demonstrated high
correlations of measured LDSA concentrations with black carbon (BC) and nitrogen oxide ($NO_x$) under certain circumstances.
Traffic activities have been observed to be significant source contribution to the LDSA concentrations (Järvinen et al., 2015).
A clear correlation was also found between the emission factors of exhaust plume BC and LDSA in on-road studies for city
buses (e.g. Järvinen et al., 2019). These highly correlating relationships provide good grounds for estimating LDSA
concentrations and short-term trends by the other pollutants measured at the same site with the use of data mining-based
approach as statistical models. Data mining-based approach exploits statistical or machine learning techniques to detect
patterns between predictors and dependent variables in the time series data. They do not demand in-depth understanding of air
pollutant dynamics, but evaluation by experts is still required to determine whether the models work properly. Simple yet
apprehensible models, such as multiple linear regression (MLR, e.g. Fernández-Guisuraga et al., 2016) and generalized
additive models (GAM, e.g. Chen et al., 2019), are commonly utilised as white-box models in air pollutant proxy studies.
Furthermore, more sophisticated machine learning black-box models, such as artificial neural network (ANN, e.g. Cabaneros
et al., 2019; Zaidan et al., 2019), nonlinear autoregressive network with exogenous inputs (NARX, Zaidan et al., 2020) and
support vector regression (SVR, e.g. Fung et al., 2021), have been intensively investigated in recent years. They work better
in terms of accuracy; however, they provide limited transparency and accountability regarding the outcomes (Rudin, 2019;
Fung et al., 2021).
Apart from model structures, the criteria of selecting variables in multipollutant datasets for model development have received
considerable attention over the years, and a large number of methods have been proposed (Miller, 2002). Traditional methods,
like stepwise procedures, which is a combination of forward selection and backward elimination (e.g. Liu et al., 2018b; Chen
et al., 2019), can be unstable because it uses restricted search through the space of potential models, which eventually causes
inherent problem of multiple hypothesis testing (Breiman, 1996; Faraway, 2014). Another approach named regularization has



emerged as a successful method to reduce the data dimension in an automated way, yet deal poorly with multi-collinear
variables, for example Least Absolute Shrinkage and Selection Operator (LASSO, e.g. Fung et al., 2021; Šimić et al., 2020),
ridge regression (e.g. Chen et al., 2019) and ELASTINET (e.g. Chen et al., 2019). Criterion-based procedures, which choose
the best predictor variables according to some criteria (e.g. coefficient of determination, residual, etc), are sensitive to outliers
and influential points, but involve a wider search and compare models in a preferable manner. Examples are best subset
regression (e.g. Chen et al., 2019), input adaptive proxy (e.g. Fung et al., 2020; Fung et al., 2021), etc. Hastie et al. (2020)
compared some of the models using the three approaches and concluded that no single feature selection method uniformly
outweighs the others. Despite the extensive research of feature selection methods, the inclusion of random effects together
with the fixed effects as linear mixed-effects (LME) model has received little attention (e.g. Font et al., 2019; Tong et al.,
2020) in air pollution research, let alone LDSA study in particular. This inclusion of random effects could acknowledge a
possible effect coming from a factor where specific and fixed values are not of interest.
In this study, we combine the use of criterion-based feature selection method and the inclusion of random effects, and develop
a novel input-adaptive mixed effects (IAME) model to estimate alveolar LDSA concentrations, which is the first study of this
context to our best knowledge. The description of LDSA measurements and the techniques of IAME model are outlined in
Sect. 2 and 3, respectively. Section 4 presents the characteristics of alveolar LDSA, including its seasonal variability, weekend
effect and diurnal pattern, in four types of environments. We also aim to investigate the correlation with other air pollutants.
In Sect. 5, we evaluate the performance of the IAME proxy ($LDSA_{IAME}$) with the measured alveolar LDSA by Pegasor AQ
Urban ($LDSA_{Pegasor}$), ICRP lung deposition model derived LDSA ($LDSA_{ICRP}$) and another modelled alveolar LDSA by IAP
($LDSA_{IAP}$) as well as the benefits and implication of this alveolar LDSA model. It should be noted that this study discusses
LDSA in alveolar region, unless stated otherwise.

## 2 Measurement description

### 2.1 Measurement sites

We retrieved aerosol, gaseous and meteorological data from two types of measurement sites, i.e., street canyon (SC, 2017–
2018) and urban background (UB, 2017–May 2018), in Helsinki Metropolitan Area (HMA) described in more detail below.
Data from detached housing (DH, 2017) and regional background (RB, 2017) sites were also included in the study to provide
comparison and data from the background concentrations. Situated on a relatively flat land at the coast of Gulf of Finland,
HMA has land area of 715 km$^2$ and population of about 1.13 million inhabitants. Helsinki can be classified as continental or
marine climate depending on the air flows and the pressure system. Figure S1 and Table S1 show the detailed site description.
**Street canyon site (SC)**: Mäkelänkatu urban supersite is operated by the Helsinki Region Environmental Services Authority
(HSY, Kuuluvainen et al., 2018). The station is located at 3 km from the city centre in a street canyon in the immediate vicinity
to one of the main roads leading to downtown Helsinki. The street, with speed limit of 50 km h$^{-1}$, consists of six lanes and two
tramlines. The annual mean traffic volume in 2018 per workday was 28 100 vehicles, 11% of which were recorded as the
heavy duty vehicles. The traffic loads are especially high during rush hours at 8 a.m. and 5 p.m. (Figure S2). The street canyon
of width of 42 m is surrounded by rows of buildings of 17 m high, which weaken the dispersion process of the direct vehicular
emissions. All the inlets for the measuring devices are positioned approximately at a height of 4 m from the ground level.
**Urban background site (UB)**: The Station for Measuring Ecosystem-Atmosphere Relations III (SMEAR III, Järvi et al.,
2009) in Kumpula, situated on a rocky hill at 26 m above sea level, is about 4 km northeast from the Helsinki centre. The
surroundings of this urban background station are heterogeneous, constituting of residential buildings, small roads, parking
lots, patchy forest and low vegetation from different direction. One main road (45 000 vehicles per workday) is located at the
distance of 150 m east from the site. Trace gases and meteorological conditions are measured at a height of 4 m and 32 m,



respectively, at a triangular lattice tower while aerosol measurements are conducted inside a container approximately 4 m
above the ground. The site is co-operated by Finnish Meteorological Institute (FMI) and the University of Helsinki (UHEL).
**Detached housing site (DH)**: Three measurement stations, Rekola (DH1), Itä-Hakkila (DH2) and Hiekkaharju (DH3), were
chosen since they represent a sub-urban residential area surrounded by detached houses. These sites are mainly affected by the
wood combustion emissions from residential activities, especially in cold weather conditions. Emissions from traffic source
also account for a small portion of the whole pollution. It is estimated that 90 % of the households burn wood to warm up
houses and saunas, less than 2 % of which use wood burning as the main heating source in detached houses in HMA (Hellén
et al., 2017).
**Regional background site (RB)**: The RB site is located about 23 km away from the Helsinki city centre at Luukki, surrounded
by a wooded outdoor recreational area right at the edge of the Greater Helsinki golf course. The measuring station is in an
open place away from busy traffic routes and large point sources. As a result, this site can represent background concentration
levels outside the urban area without any main local sources.

**2.2 Instruments**

**LDSA measurements:** The sensor unit and the core of the Pegasor AQ Urban is practically another instrument called a Pegasor
PPS-M sensor (Pegasor Ltd., Finland) originally designed for automotive exhaust emission measurements (e.g. Maricq, 2013;
Amanatidis et al., 2017). The operation of the sensor is based on diffusion charging of particles and the measurement of electric
current without the collection of particles. The diffusion charging of particles is carried out by a corona-ionized flow that is
mixed with the ambient sample air in an ejector diluter inside the sensor. The sampling lines and the sensor unit are heated
40°C above the ambient temperature (1) to dry the aerosol sample, (2) to prevent interference from humidity, and (3) to prevent
any water condensation inside the sensor. The performance of the Pegasor PPS-M sensors for long-term ambient measurements
has been improved after they were tested in Helsinki (Järvinen et al., 2015) and Beijing (Dal Maso et al., 2016). The suggestions
have been considered for the design of the current form of the Pegasor AQ Urban in this study.
The Pegasor AQ Urban (dimension: 320 mm×250 mm×1000 mm), which consists of a weatherproof cover, clean air supply,
and the abovementioned Pegasor PPS-M sensor, has been designed such that its response to LDSA is not to be subjected to
meteorological fluctuation for outdoor operation. Kuuluvainen et al. (2016) used two Pegasor AQ Urban devices during a 2
week period at an urban street canyon and an urban background measurement station in Helsinki, Finland whereas Kuula et
al. (2019) later used instrument in a 3 month long campaign at the same urban street canyon station. These studies demonstrated
that the output signal of the Pegasor AQ Urban correlated well with other devices measuring LDSA concentrations such as the
Partector and DiSCmini. Kuula et al. (2020) further validated the accuracy and stability of Pegasor AQ Urban at the street
canyon station by comparing the measured values of one full year with DMPS reference instruments ($R^2 = 0.90$, RMSE = 4.1
$\mu m^2 \, cm^{-3}$). The instrument is optimized to measure the alveolar LDSA concentrations of particles in ~10–400 nm size range.
Pegasor AQ Urban tends to underestimate LDSA of particle larger than about 400 nm. In typical urban environments, most of
the particles from local combustion sources are in the size below the threshold (Kuuluvainen et al., 2016; Pirjola et al., 2017).
However, the impact of larger particles (>400 nm) to alveolar LDSA might be significant, for example a recent study on LDSA
concentrations in polluted urban environment in India observed high LDSA contribution from relatively large accumulation
mode particles although the experiment was conducted in close proximity of traffic (Salo et al., 2021a) and in mining
environment the mineral dust and other pollutants being typically in larger particle sizes can also contribute to the LDSA
concentrations (Salo et al., 2021b). In HMA, the impact of >400 nm might also be significant during $PM_{2.5}$ long-range transport
episodes or when there are many particles from very low-quality residential burning in detached housing areas (Pirjola et al.,

209  2017).

**Aerosol measurements:** Differential mobility particle sizer (DMPS) in combination of a differential mobility analyser (DMA)
and a condensation particle counter (CPC) measures aerosol size distribution (Kulkarni et al., 2011). Vienna DMA and



Airmodus A20 CPC (measurements of particle size range 6–800 nm) are used at the SC site while a twin DMPS (Hauke-type
DMA and TSI Model 3025 CPC + Hauke-type DMA and TSI Model 3010 CPC, merged particle size range 3–1000 nm) are
used at the UB site. Both instruments make use of the bipolar charging of aerosol particles, followed by classification of
particles into size classes according to their electrical equivalent mobility. In addition to particle size distribution, total particle
number concentration (PNC, in cm$^{-3}$) is calculated by summation. Particle mass concentration of diameter less than 2.5 µm
(PM$_{2.5}$, in µg m$^{-3}$) and less than 10 µm (PM$_{10}$, in µg m$^{-3}$) are measured continuously with ambient particulate monitor TEOM
1405 at the SC site and TEOM 1405-D at the UB site. Black carbon (BC, in µg m$^{-3}$) mass concentration is measured by a
multi-angle absorption photometer (MAAP) Thermo Scientific 5012 with a PM$_1$ inlet. The measured absorbance is converted
to BC mass concentration by using a fixed 6.6 m$^2$ g$^{-1}$ mass absorption coefficient at wavelength of 637 nm. PM$_{2.5}$, PM$_{10}$ and
BC are recorded in µg m$^{-3}$.
**Ancillary measurements:** Trace gas concentrations (in ppb), including nitrogen oxide (NO), nitrogen dioxide (NO$_2$), their
sum nitrogen oxide (NO$_x$), ozone (O$_3$) and carbon monoxide (CO) are determined with a suite of gas analysers. In addition,
supporting meteorological variables, including air temperature (Temp), relative humidity (RH), air pressure (P), wind speed
(WS), wind direction (WD) and photosynthetically active radiation (PAR), are measured at SC and UB. Figure S3 show the
meteorological conditions during the measurement period. A list of variables collected is shown in Table S2.
**3. Method**
**3.1 Data pre-processing**
The collected data was quality checked by the corresponding operating organisation, HSY, FMI and UHEL. No additional
pre-processing was done for general analysis. For proxy development, outliers were detected using the interquartile range
(IQR) rule, which is applicable for non-Gaussian distribution sample. We calculated the cut-off for outliers as 2 times the IQR,
subtracted this cut-off from the 25$^{th}$ percentile and added it to the 75$^{th}$ percentile to give the actual limits on the data. We
applied a natural logarithm transformation to all the skewed-distributed aerosol and trace gases measurements in order to keep
the distribution of each parameter following a normal distribution. Since wind direction is a circular variable, it is resolved
into North-South (WD–N) and East-West (WD–E) vector components by trigonometric functions.
**3.2 Size-fractionated lung deposited surface area (LDSA$_{ICRP}$)**
Alveolar deposition fraction ($DF_{AL}$) as a function of particle size with the unit density is determined with the ICRP Human
Respiratory Tract Model by the following equation (ICRP, 1994).

$$DF_{AL} = \left(\frac{0.0155}{d_p}\right)\left(\exp\left(-0.416\left(\ln d_p + 2.84\right)^2\right) + 19.11 \exp\left(-0.482\left(\ln d_p - 1.362\right)^2\right)\right) \quad \textbf{(1)},$$

where $d_p$ is the aerodynamic diameter (µm) of spherical particles with the unit density (1 g cm$^{-3}$). The equation is determined
in two parts with respect to the two different peaks in the deposition curve in Figure 1. The peak near the size of 20 nm can be
approximated to represent the Brownian deposition, whereas the peak between 1 µm and 2 µm represents the inertial
deposition. From the particle number size distribution, we calculated the particle surface area distribution assuming each
particle is monodisperse sphere of standard density at standard conditions. By Eq. (1), a deposition factor for each particle size
bin (26 size bins at SC and 49 at UB) were calculated. Size-fractionated LDSA was then computed by multiplying the surface
area concentration with $DF_{AL}$ in the corresponding size class. Total LDSA calculated by the ICRP lung model (LDSA$_{ICRP}$) can
be obtained by summing up the all the size-fractionated LDSA values. In this study, the alveolar LDSA$_{ICRP}$ was calculated
based on DMPS measurements in SC and UB. Thus, while the alveolar LDSA measured by Pegasor (LDSA$_{Pegasor}$) represent
the ~10–400 nm size range, the alveolar LDSA$_{ICRP}$ represent 6–800 nm and 3–1000 nm size range in SC and UB, respectively.





**3.3 Novel Input-adaptive mixed-effects (IAME) model**
Input-adaptive mixed-effects (IAME) model is a combination of input-adaptive proxy (IAP) and linear mixed-effects (LME)
model. IAP was first introduced by Fung et al. (2020) and has been demonstrated reliable and flexible to fill up missing values
by taking input variables adaptively with robust ordinary least square regression models. IAP has been able to estimate BC
concentration by other air quality indicators with a satisfactory performance in two different categorised urban environments,
street canyon (adjusted $R^2 = 0.86$–$0.94$) and urban background (adjusted $R^2 = 0.74$–$0.91$). Some models outperformed IAP in
accuracy performance, but its transparent model structure and ability to impute missing values still make it a preferred option
as a virtual sensor (Fung et al., 2021).

In this study, we primarily stick to the strength to select input variables adaptively with the introduction of mixed effects. The
mixed effect approach is a generalization of the linear model that can incorporate both fixed (i.e. causing a main
effect/interaction) and random effects (i.e. causing variance/variability in responses), allowing the account of several sources
of variations (Chudnovsky et al., 2012). As seen in Figure 2, We picked the direct air pollutant measurement from the station
(variables of high correlation: $PM_{2.5}$, BC and $NO_2$ and other supporting variables: $PM_{10}$, $O_3$, $NO_x$, NO, CO and PNC) and
meteorological data of higher correlation (Temp, RH, P, PAR, WS, WD–N, WD–E) as the fixed variables because the air
pollutants can indicate the sources of LDSA which largely come from combustion and meteorological data could influence the
dispersion and dilution of LDSA. They are the most direct factors to the fluctuation of LDSA concentrations. Due to the strong
seasonal variation, weekend effects and diurnal pattern in urban air pollutant concentrations (Fung et al., 2020), the variance
in responses might depend on the time indicators that are not the primary cause of the concentration variability, but they
indirectly alter human-induced activities, such as traffic amounts. To take them into account, we created three time hierarchical
sub-groups (12 months of year, 7 days of week and 24 hours of day) as the inputs of random effect variables.

The regression equation of IAME is similar to the equation of IAP, except that IAME includes additional intercepts term for
random effects as below:

$$y_i = \beta_0 + \sum_{k=1}^{p} \beta_{ik} x_{ik} + \sum_{j=1}^{q} b_{ij} + e_i$$

(2),

where $y_i$ is the $i^{th}$ estimated LDSA concentration. The first term on the right $\beta_0$ indicates the fixed intercept of the equation.
The second term represents the total contribution by the direct measurement of variable $x$ as fixed effects with a slope $\beta$ at
each data point $i$. A maximum of three inputs from the total 16 fixed variables are selected to from 696 sub-models (Figure 2).
The inputs for random effects are indicated by $b$ as intercepts of the corresponding three hierarchical sub-groups. A Gaussian
error term is indicated by $e$. The explanation of Eq (2), is visualised in Figure 2.

One of the assumptions of LME models is that the random effects, together with the error term, have the following prior
distribution:

$$b \sim N\big(0, \sigma^2 D(\theta)\big)$$

(3),

where $D$ is a $q$-by-$q$ symmetric and positive semidefinite matrix, parameterized by a variance component vector $\theta$, $q$ is the
number of variables in the random-effects term, and $\sigma^2$ is the observation error variance. We use an optimiser, restricted
maximum likelihood, commonly known as ReML, with the value $1 \times 10^{-6}$ as the relative tolerance on gradient of objective
function and $1 \times 10^{-12}$ as absolute tolerance on step size. The use of ReML over the conventional ML could produce unbiased
estimates of variance and covariance parameters (Lindstrom and Bates, 1988).





After the sub-model formation, the dataset is randomly divided into five portions. 80% of the data are allocated for 4-fold cross
validation to remove variance of accuracy. The results of all the folds are averaged and the sub-models are ranked by several
evaluation metrics, which are further demonstrated in Figure 2 and described in Sect. 3.4. Some of the sub-models are subject
to rejection under two conditions: (1) strong multi-collinearity among the fixed parameters (variance inflation factor (VIF)
exceeding a threshold of 5) and (2) violation of the normality assumption of residuals also known as heteroscedasticity (fail in
Kolmogorov-Smirnov (K-S) test, $p < 0.05$). Based on the situation of missing data, the automatised IAME model will search
for the best sub-model option from the ranking chart. Hence, each data point might be estimated differently depending on the
available data. The number of data points being estimated by each sub-model is reported to show their frequency of usage.

**3.4 Evaluation metrics**

In order to evaluate the model performance quantitatively, we use the following metrics:

$$R^2 = 1 - \frac{\sum_{i=1}^{N}(y_i - \hat{y}_i)^2}{\sum_{i=1}^{N}(y_i - \bar{y})^2} \tag{4},$$

$$MAE = \frac{1}{N}\sum_{i=1}^{N}|y_i - \hat{y}_i| \tag{5},$$

$$cRMSD = \sqrt{\frac{1}{N}\sum_{i=1}^{N}((y_i - \bar{y}) - (\hat{y}_i - \tilde{y}))^2} \tag{6},$$

$$r = \frac{\sum_{i=1}^{N}(y_i - \bar{y})(\hat{y}_i - \tilde{y})}{\sqrt{\sum_{i=1}^{N}(y_i - \bar{y})^2}\sqrt{\sum_{i=1}^{N}(\hat{y}_i - \tilde{y})^2}} \tag{7},$$

$$NSD = \frac{SD_{predicted}}{SD_{reference}} = \frac{\sqrt{\frac{1}{N-1}\sum_{i=1}^{N}(\hat{y}_i - \tilde{y})^2}}{\sqrt{\frac{1}{N-1}\sum_{i=1}^{N}(y_i - \bar{y})^2}} = \sqrt{\frac{\sum_{i=1}^{N}(\hat{y}_i - \tilde{y})^2}{\sum_{i=1}^{N}(y_i - \bar{y})^2}} \tag{8},$$

where $\hat{y}_i$ and $\hat{y}_i$ are $i^{th}$ measured data point and estimated variable by the model, respectively. $\bar{y}$ and $\tilde{y}$ are the expected value
of the measured and modelled dataset, respectively. $N$ is the number of complete data input to the model. Coefficient of
determination ($R^2$) is a measure of how close the data lie to the fitted regression line. It, however, does not consider the biases
in the estimation. Therefore, we further validated the models with mean absolute error ($MAE$) and centred root-mean-square
differences ($cRMSD$), where $MAE$ measures the arithmetic mean of the absolute differences between the members of each
pair, whilst $cRMSD$ calculates the square root of the average squared difference between the forecast and the observation pairs.
$cRMSD$ is more sensitive to larger errors than $MAE$. Furthermore, together with $cRMSD$, Pearson correlation coefficient ($r$)
and normalised standard deviation ($NSD$) of the modelled data set are also studied. $r$ describes the correlation between the
measured and modelled data whereas $NSD$ measures the relative spread of the data. Due to their unique mathematical
relationship, the three metrics can be portrayed on Taylor's diagram, which has been used for sub-model selection purpose.
We ranked our sub-models first by $R^2$, followed by $MAE$ and $cRMSD$. $r$ and $NSD$ serve as additional evidence when we
explain the model performance.

**3.5 Two-sample t-tests**

We assessed the temporal and spatial impact on the IAME model by comparing the means of absolute differences between the
hourly measured and modelled LDSA in different time windows at both stations. Two-sample t-tests were performed on the
two populations of absolute differences abovementioned to determine whether the difference between these is statistically
significant. A significance level α of 5% is chosen as the probability of rejecting the null hypothesis when it is true, denoted
as $p$.



### 4 LDSA measurement characterization

#### 4.1 General characteristics of LDSA$_{Pegasor}$ in Helsinki metropolitan area

The annual mean alveolar LDSA concentrations at four station types SC (2017–2018), UB (2017–May 2018), DH (2018) and RB (2018) are 19.7±11.3 µm$^2$ cm$^{-3}$, 11.2±7.1 µg m$^{-3}$, 11.7±8.6 µm$^2$ cm$^{-3}$ and 7.6±5.4 µm$^2$ cm$^{-3}$, respectively (Table 2). The DH and RB site are included to give more substantial interpretation of data because the LDSA concentrations at RB can be viewed as background measurements and the local LDSA increments in HMA can be represented by the LDSA at the hotspot measurement site subtracted by the LDSA at the RB site. The timeseries of LDSA concentrations at the SC and the UB site are presented in Figure 3 and Fig. S4, where the missing data of LDSA for the whole measurement period is 3% and 30%, respectively. When comparing with the same site type in other cities around the globe, LDSA concentrations detected in HMA are the lowest among the European cities with reported values, and about one-fifth that in Japan (**Table 1**). Some literatures also report LDSA at tracheobronchial region but most just consider LDSA at alveolar which is considered to bring most harm to human's lungs.

The diurnal pattern of LDSA at RB is not observable on workdays or over weekends (Figure 4, upper panel). The relatively low variability can be explained by the scarcity of human activities. We can then regard the LDSA at RB as the background concentrations mainly influenced by the regionally and long-range transported aerosol and meteorological variation. As the concentrations at RB is stable throughout the different hours of day; therefore, the diurnal pattern of LDSA concentration is apparently indistinguishable between the measured concentration and the local increments. At the UB and DH site, the magnitudes and the patterns of the average hourly LDSA concentrations at workdays are comparable, and both show bimodal curves, one peak at 6−9 a.m., the other at 9−11 p.m.. The former has a larger peak during the morning peak hour because of the vehicular emissions (Timonen et al., 2013; Teinilä et al., 2019) while the latter has a larger peak in the evening attributed mainly by the residential burning (Hellén et al., 2017; Helin et al., 2018; Luoma et al., 2021). Over weekends, the peaks in the morning are not identifiable and the evening peaks are amplified due to enhanced human activities. Similar diurnal variation at residential area was observed for BC emitted by residential combustion by Helin et al. (2018). At the SC site, the morning peak on weekends is not obvious because of the lack of work-related traffic. It appears that a similar bimodal curve can be seen during workdays, but the evening peak is seen during the evening traffic rush hour around 4−6 p.m.. The reason is that the main contributor of LDSA at the SC site is traffic and combustion processes and the diurnal variability mainly depends on the citizen's movement by vehicles in the city. Over weekends, the average hourly LDSA concentrations are the minimum at 5 a.m. and they increase and remain at a high level at 2 p.m. until the late night. The level of LDSA concentrations at DH is comparable with that at UB site. However, the amplitudes of the evening peak is higher than that of the morning peak both on workdays and weekends due to elevated residential combustion.

However, the monthly variability of background measurements at the RB site is stronger compared to the diurnal pattern and the calculation of local increment is necessary. With no intense point sources, the variations at RB are probably due to horizontal dispersion and advection of aerosol particles and vertical dilution controlled by the boundary layer dynamics. In the summer, when solar radiation is persistently stronger, the boundary layer becomes elevated due to surface heating and associated thermal turbulence. This turbulence would dilute the concentration of pollutants at the surface. Another plausible reason could be the higher regional and long-range transported LDSA in the summer, as demonstrated by Kuula et al. (2020) and Barreira et al. (2021). The lower panel in Figure 4 shows the LDSA local increments after subtraction of the LDSA at the RB site. For instance, the local LDSA increments at DH are the highest in the winter probably due to local small-scale wood combustion (and traffic). However, without subtracting the background concentrations, the LDSA concentrations at DH are higher in the summer than in the winter (due to high regional background concentrations in summer), as was observed also by



Kuula et al. (2020). This piece of evidence can help in the source apportionment. The variation of diurnal and seasonal LDSA
for all sites are visualised in Fig. S5.
**4.2 The connection between LDSA and other parameters**
Alveolar LDSA concentration, as a single number, comprises particles across the whole particle size spectrum measured (e.g.
Pegasor AQ Urban ~10–400 nm). InHMA, the two local main sources of particles contributing to LDSA are vehicular
combustion and residential wood combustion emissions. Upon the two combustion processes, particles of different sizes and
different gaseous pollutants are emitted. A study by Lamberg et al. (2011) has shown that the geometric mean diameter of
residential wood combustion is typically 70–150 nm whereas Barreira et al. (2021) presented that the typical particle size for
vehicular combustion can be smaller than 50 nm. By calculating the proportion of LDSA with respect to different pollutant
parameters BC, $NO_x$, PNC (dominated by UFP), and $PM_{2.5}$, we could identify the contribution of LDSA across the hour of day
(Fig. S6 for workdays and Fig. S7 for weekends). Since the vehicular combustion emits smaller particles which elevate the
LDSA concentration but meanwhile do not substantially influence the value of $PM_{2.5}$ (e.g. Salo et al., 2021a); therefore,
LDSA/$PM_{2.5}$ has a diurnal pattern similar to the LDSA concentrations which peaks in the morning rush hour during workdays.
Conversely, LDSA/BC, LDSA/PNC and LDSA/$NO_x$ have a higher value before the morning rush hour and they plunged in
the morning rush hour. This can be explained by the fact that vehicular combustion emits high concentration of BC, PNC and
$NO_x$ (Reche et al., 2015) compared to its contribution to LDSA concentration. In other words, the role of regional background
is higher for LDSA compared to those of $NO_x$, BC and PNC. At the UB site, the average LDSA/BC at all hours remain at a
constant level in the winter while the variability of the ratio is much higher in the summer. The general LDSA/PNC ratio at
UB is steadily 2−3 times higher than that at all hours in all seasons because the proportion of larger particles at UB is usually
higher than SC. This large variability again validate the heterogeneity of source of LDSA.

The integrated alveolar LDSA with a various size ranges was calculated to explore the correlation of size-fractionated LDSA
and other parameters in our multipollutant dataset. No single fractionated LDSA correlates well with meteorological
parameters at both sites (Figure 5). Out of all fractions, alveolar LDSA of the whole spectrum ($LDSA_{6–800}$) and $LDSA_{250–400}$,
which explains majority of LDSA, correlates best with other air pollutants. In general, alveolar LDSA has high correlation
with BC. BC correlates the best with $LDSA_{100–250}$, which is in alignment with the reported values from previous literature
(Gramsch et al., 2014; Ding et al., 2016). As expected, $PM_{2.5}$ show better correlation with the LDSA of larger particles because
larger particles contributes more to $PM_{2.5}$ mass concentration values. In the meanwhile, $PM_{10}$ has fair correlation with all
selected size bins. $NO_2$ correlates highly with LDSA of smaller particles, indicating the dominant role of local traffic exhausts.
CO has a higher correlation with LDSA of 400−800 nm since CO concentrations are more affected by regionally transported
pollutants. $O_3$ has a fair correlation with LDSA of all sections because the formation of $O_3$ is mostly secondary and the chemical
interactions with pollutants are more complicated than the other compounds. In general, the correlation of LDSA with other
air pollutant parameters is higher at the SC site than that at the UB site (Fig. S8). The high correlation of LDSA with BC, $PM_{2.5}$
and $NO_2$, which agrees with the results by Kuula et al. (2020), proves the possibility of developing a model to estimate LDSA
concentrations.
**5 Model evaluation**
**5.1 Sub-model diagnostics**
Following the evaluation attributes described in Sect. 3.4,





Table 3 depicts the descriptive statistics of the overall model evaluation on its testing set. The overall model at the SC site is
able to explain 80% of the variability of the testing set of the measured data. The $R^2$ in the winter is 0.86 being the highest
while the worst $R^2$ is shown in the summer, i.e., 0.70. The $MAE$ and $cRMSD$ are the smallest during weekend with $R^2$ not
particularly high ($R^2 = 0.72$) probably because the LDSA concentration itself is relatively low in that period. The overall
performance is generally worse in UB in terms of $R^2$, except during weekends that $R^2$ is 10% higher.

For individual sub-models, their performance could be seen on the Taylor's diagram in Figure 6 (Taylor, 2001). Each marker
represents one sub-model, the contribution of which to the outcome of the final model is displayed in various colours. The
sub-model performance can be evaluated by the distance of the sub-model marker and the red point, which represents the
reference station, i.e., the perfect model. The location of each marker indicates its individual performance in terms of $r$,
$cRMSD$ and $NSD$. At the SC site, the narrow distribution of the sub-models on the Taylor's diagram gives a clue that they are
very similar in terms of model performance of LDSA estimation. The five mostly used sub-models are concentrated within
the region where $r$ is 0.85–0.87, $cRMSD$ is 5.67−5.77 µm² cm⁻³ and $NSD$ is 0.75−0.79 (Table 4). The values of their
evaluation metrics are close to each other where R² and $MAE$ differ in the narrow range of 10% ($R^2 = 0.72$–0.74, $MAE = 3.8$
µm² cm⁻³). It infers that if one metric is prioritised over another, the rank of the sub-models can be greatly different. Although
no individual sub-models show $r$ greater than 0.9, the overall model comprising the outcomes by all the sub-models remains
high ($R^2 = 0.80$, $MAE = 3.8$ µm² cm⁻³). The best sub-model is also the most used one, which accounts for 81% of the total
data points while the two succeeding sub-models constitute another 16%. This also indicates that the input adaptivity function
of the suggested method supplement 19% of estimates which would be a missing estimate if a single model with fixed predictor
variables is used. Four out of the five most used sub-models contain BC as an input predictor with the combination of other
two air pollutants or meteorological parameters. In case BC is missing at a certain time stamp, the sub-model without BC as
an input could be used. It further supports the input adaptive function.

At the UB site, the sub-model performance is more scattered on the Taylor's diagram (Figure 6). The five most used sub-
models have varying metrics ($r = 0.77−0.92$, $cRMSD = 2.5−3.9$ µm² cm⁻³ and $NSD = 0.63−0.89$, see Table 5). Although some
show exceptionally good performance, the overall model has a slightly worse performance than that in street canyon. The best
sub-model estimates 49% of the total measurement, followed by 17%. The third and fourth most used sub-models, which form
up to 30% of the estimates, have rather moderate performance ($R^2 = 0.58$ and 0.69). Considering all possible outcomes, the
overall model is still able to explain 77% of the total variance. CO and PNC dominate in the top five used sub-models. BC,
NOₓ and meteorological parameters, like RH and WD-N are also involved in the final LDSA estimation.

By checking the variance inflation factor (VIF) of all 696 sub-models, 4.6% and 2.2% are rejected respectively. The higher
rejection rate at SC can be explained by the fact that some of the predictor variables are highly correlating to each other and
the inclusion of them would result in an inflation of multi-collinearity of the sub-model, from which biases arise. At UB, since
the source of LDSA is more varied and the correlation of LDSA with other pollutants is generally lower, the probability of the
VIF of the individual sub-models exceeding the threshold is lower.
**5.2 Temporal difference in comparison with other models**
Figure 7 presents the comparison of measured LDSA (LDSA_Pegasor), deposition model derived LDSA (LDSA_ICRP) and the
LDSA modelled by IAP and IAME (LDSA_IAP and LDSA_IAME) as a timeseries plot between 14 and 28 February 2017. This
particular time window is selected because it suffers the least in data missing for all the respective instruments at both sites.
This figure during this period can also showcase the difference in magnitudes of the diurnal shape over workdays and weekends
(shaded regions in Figure 7). At the SC site, the estimates by both LDSA_IAP and LDSA_IAME could generally catch up with the





diurnal cycle of the measured data. However, the models underestimate the peak if the change of the measured LDSA
concentration is sudden and relatively large. Despite the small difference observed in the figure, the blue dotted line
representing LDSA$_{IAME}$ often stays closer to the measured LDSA concentration (black line). When we smooth out all the
estimates at each hour, the ability for IAME to catch the morning peak on workdays is much better. At the UB site, IAME
underestimates the LDSA concentration by almost 50% and 25% in the morning on 15 and 23 February 2017, respectively.
The overestimation reaches 100% during the midnight between 26 and 17 February 2017.

A more generalised diurnal cycle can be found in Figure 8. The error bars of the modelled LDSA$_{IAP}$ and LDSA$_{IAME}$ are
consistently smaller than that of LDSA$_{Pegasor}$ and LDSA$_{ICRP}$. It might be due to the reason that the model fails to catch the
extreme values although it manages to catch the general diurnal cycle. Since outliers are removed in the pre-processing stage
and the model penalises the extreme values, the model tends to give a more centralised estimate. It is a trade-off between the
option with better coefficients of determination but stronger extreme errors and that with better estimations at tails but
derivation of averaged estimation. This circumstance is more apparent on workdays than weekends. Furthermore, LDSA$_{IAME}$
could follow the diurnal cycle of LDSA$_{Pegasor}$ much better than LDSA$_{IAP}$, especially during the start of the peak hours over
workdays at the SC site where the LDSA concentrations jump to a high level. LDSA$_{IAME}$ can explain 80% and 77% of the
variability of the reference measurements at SC and UB, respectively (Table 6





Table 6), and compared to LDSA$_{IAP}$'s 77% and 66%, LDSA$_{IAME}$ perform better in terms of accuracy. In addition, the slightly
smaller $MAE$ and the closer to 1 $NSD$ of the LDSA$_{IAME}$ suggest that the mean absolute error is improved and the spread of the
estimation distribution is closer to the reference measurement by taking random effects into account.

Furthermore, we assessed the temporal and spatial impact on the IAME model by comparing the means of absolute differences
between the hourly LDSA$_{Pegasor}$ and LDSA$_{IAME}$ in different time windows at both stations. A descriptive statistic is presented
in Table 7. We used two-sample t-tests to assess whether the distribution of absolute differences were statistically significant.
At SC, the $p$ value of the t-tests at all selected windows are below 0.05, which demonstrate that the performance at different
seasons, days of week and hours of day of absolute differences between the measured and modelled LDSA were significantly
different at the confidential level of 95%. At the UB site, the difference between the two selected hour periods is not statistically
significant. The same applies to the difference between winter and spring. There are no statistically sufficient evidence to
validate the difference among the rest of the selected time period. In other words, with the use of random effects of time
constraint, the overall models still perform differently at different time windows most of the time. This indicates that IAME
still needs improvements on minimising temporal differences.
**6 Conclusion**
In this study, we develop a novel input-adaptive mixed-effects (IAME) proxy, to estimate alveolar LDSA by other already
existing air pollutant variables and meteorological conditions in Helsinki Metropolitan Area. During the measurement period
2017–2018, we retrieved LDSA measurements measured by Pegasor AQ Urban (alveolar LDSA in the ~10−400 size range)
and other variables in a street canyon (SC, average LDSA = 19.7±11.3 µm² cm⁻³) site and an urban background (UB, average
LDSA = 11.2±7.1 µm² cm⁻³) site in Helsinki, Finland. Furthermore, three detached housing sites (DH, average LDSA =
11.7±8.6 µm² cm⁻³) and a regional background site (RB, average LDSA = 7.6±5.4 µm² cm⁻³) are also included as reference
and background source estimation, respectively. At the SC site, LDSA concentrations are closely correlated with traffic
emission. The ratio to black carbon (LDSA/BC), to particle number concentration (LDSA/PNC), and to nitrogen oxide
(LDSA/NO$_x$) have a higher value before the morning peak and it reaches its minimum during the morning peak since the role
of regional background is higher for LDSA compared to those of NO$_x$, BC and PNC. However, the ratio of LDSA to mass
concentration of particles of diameter smaller than 2.5 µm (LDSA/PM$_{2.5}$) perform differently since the freshly vehicular
emitted particles are smaller than 50 nm, which do not contribute much to PM$_{2.5}$ mass concentration.

For the continuous estimation of LDSA, IAME is automatised to select the best combination of input variables, including a
maximum of three fixed effect variables and three time indictors as random effect variables. Altogether, 696 sub-models are
generated and ranked by the coefficient of determination ($R^2$), mean absolute error ($MAE$) and centred root-mean-square
differences ($cRMSD$) in order. At the SC site, LDSA concentrations can be best estimated by PM$_{2.5}$, PNC and BC, all of which
are closely connected with the vehicular emissions, while they are found correlating with PM$_{2.5}$, BC and carbon monoxide
(CO) the best at the UB site. At both sites, PM$_{2.5}$ also indicates the regionally and long-range transported pollutants, which is
a significant source of LDSA concentrations. The accuracy of the overall model is higher at the SC site ($R^2 = 0.80$, $MAE =$
3.7 µm² cm⁻³) than at the UB site ($R^2 = 0.77$, $MAE = 2.3$ µm² cm⁻³) plausibly because the LDSA source was more tightly
controlled by the close-by vehicular emission source. This model could catch the temporal pattern of LDSA; however, the
two-sample t-tests of the residuals at all selected time windows show that their distributions are different. This indicates that
the model still performs differently at different time windows. Despite this, the novel IMAE model works better in explaining
the variability of the measurements than the previously suggested IAP model as indicted by a higher $R^2$ and lower $MAE$ in





both sites. This adjustment by taking random effects into account improves the sensitivity and the accuracy of the fixed effect
model IAP.

The models alone cannot replace the need for reference measurements. However, the IAME proxy could serve as virtual
sensors to complement the measurements at reference stations in case of missing data. The two measurement sites in this study
serve as a pilot of the proxy development, and the next step is to extend the work to the existing network of several measurement
stations within the Helsinki metropolitan region. With similar configurations, we could fill up the voids with the information
from the other stations after conscientious calibration. For example, in this paper, the two measurement sites are characterised
as street canyon and urban background. In a different setup, we may assume the similarity of the same type of environment
and utilise the measurements as replacement.

Furthermore, this continuous LDSA estimation could be useful in updating some of the current air quality application, for
instance GreenPaths application which searches for the best route to wished destination with the least exposure to air pollution
(Poom et al., 2020) and ENFUSER air quality model which provide accurate spatio-temporal estimation for air pollutants in
Helsinki (Johansson et al., 2015).

**Data availability**
The air quality data and meteorological data are available from HSY website (https://www.hsy.fi/avoindata) and through
SmartSMEAR online tool (https://smear.avaa.csc.fi/).
**Author contributions**
PLF performed formal analysis and writing – original draft of the manuscript. PLF, MAZ, TP and TH conceptualized and
designed the methodology of this work. MAZ, ST, MK, TP and TH provided supervision in this research activity. ES (Pegasor
Ltd.), JVN and AKo (HSY), and HT, JK and AKa (FMI) provided instruments and data for the campaign. All the co-authors
(MAZ, JVN, ES, HT, AKo, JK, TR, Aka, ST, MK, TP and TH) reviewed and commented on the manuscript.
**Competing interests**
Prof. Markku Kulmala and Prof. Tuukka Petäjä are members of the editorial board of the journal Atmospheric Chemistry and
Physics. Dr. Erkka Saukko works in Pegasor Ltd. which is the manufacturer of Pegasor AQ Urban.
**Acknowledgements**
The authors acknowledge the City of Helsinki for providing traffic count data.
**Financial support**
This work is supported by the European Regional Development Fund through the Urban Innovative Action (project HOPE;
Healthy Outdoor Premises for Everyone, project no. UIA03-240) and Regional Innovations and Experimentations Fund AIKO,
governed by the Helsinki Regional Council (project HAQT; Helsinki Air Quality Testbed, project no. AIKO014). Grants are
also received from the European Research Council through the European Union's Horizon 2020 Research and Innovation





Framework Program (grant agreement no. 742206), and ERA-PLANET (www.era-planet.eu) and its trans-national project

SMURBS (www.smurbs.eu) funded under the same program (grant agreement no. 689443). The authors show gratitude to

Academy of Finland for the funding via the Academy of Finland Flagship funding (project no. 337549 and 337552) and

NanoBioMass (project no. 1307537).

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

**Table 1.** Ambient LDSA of alveolar region (in µm$^2$ cm$^{-3}$, corrected to 2 significant figures) reported in the last decade in chronological
order of the measurement start. TS and RA represent traffic sites and residential area respectively. For the other acronyms, please see the
method section.

| Site description | Location | Average (Mean, unless state otherwise) | Uncertainties (SD, unless state otherwise) | Period/Season | Instruments | Study |
|---|---|---|---|---|---|---|
| UB | Ruhr, Germany | median=36 | IQR=21 | Mar 2009–Dec 2014 | NSAM | Hennig et al. (2018) |
| RB+UB+TS | Basel, Geneva, Lugano, | 32 | IQR=25 | Jan 2011–Dec 2012 | DiSCmini | Eeftens et al. (2016) |





| | Wald, Switzerland | | | | | |
|---|---|---|---|---|---|---|
| City centre with heavy traffic | Lisbon, Portugal | 35–89 | 4–8 | Apr–May 2011 | NSAM | Albuquerque et al. (2012) |
| UB | Cassino, Italy | 88–240 | - | Oct 2011– Mar 2012 | NSAM | Buonanno et al. (2012) |
| RB | | 69 | | | | |
| UB with traffic influence | Barcelona, Spain | 37 | 26 | Nov 2011–May 2013 | NSAM | Reche et al. (2015) |
| TS | Helsinki, Finland | 65–94 | - | Feb 2012 | ELPI, NSAM | Kuuluvainen et al. (2016) |
| RA | | 15–31 | | | | |
| TS | Athens, Greece | 65 | 21 4.8 | Jul 2012 | Partector Aerotrak 9000 | Cheristanidis et al. (2020) |
| UB with traffic influence | Leichester, UK | 30 | 25 | Nov 2013–May 2015 | NSAM | Hama et al. (2017) |
| | | 23 | 14 | Warm months | | |
| | | 38 | 33 | Cold months | | |
| Airport | Los Angeles | 47 | 27 | Nov–Dec 2014 and May–Jul 2015 | DiSCmini | Habre et al. (2018) |
| UB | Fukuoka, Japan | 127 | 62 | Apr 2015–Mar 2016 | NSAM | Kiriya et al. (2017) |
| TS | Helsinki, Finland | 60 (ground level) 36-40 (below rooftop) 16-26 (above rooftop) | | Nov 2016 | Partector, ELPI, DiSCmini, Pegasor AQ Urban | Kuuluvainen et al. (2018) |
| SC | Helsinki, Finland | 22 | 14 | Feb 2017–Jan 2018 | Pegasor AQ Urban | Kuula et al. (2020) |
| UB | | 9.4 | 6.9 | | | |
| DH | | 12 | 10 | | | |
| TS | Delhi, India | 330 | 130 | Nov–Dec 2018 | ELPI | Salo et al. (2021a) |
| UB | Salerno | 79 | 48 | Nov 2018– May 2019 | NanoTracer | Pacitto et al. (2020) |
| TS | Roma, Italy | 110 | 57 | | | |
| RB | Parma, Italy | 17 | 10 | | | |







**Table 2.** Descriptive statistics of alveolar LDSA concentrations ($\mu m^2\ cm^{-3}$) at SC (2017–2018), UB (2017–May 2018), DH1–3 (2018) and
RB (2018) site. The mean (column 3), standard deviation (SD, column 4), 10th, 25th, 50th, 75th and 90th percentile (P10, P25, P50, P75 and
P90, column 5–9), geometric mean (Gmean, column 10) and geometric standard deviation (GSD, column 11) of the concentrations are
corrected to one decimal place. The percentage of valid data in the reported measurement period is shown in column 12.

| | | Mean | SD | P10 | P25 | P50 | P75 | P90 | Gmean | GSD | % |
|---|---|---|---|---|---|---|---|---|---|---|---|
| SC | All | 19.7 | 11.3 | 8.4 | 11.7 | 17.0 | 24.7 | 34.4 | 17.0 | 1.7 | 97 |
| | Winter | 19.4 | 12.2 | 7.6 | 10.7 | 16.1 | 24.7 | 35.3 | 16.3 | 1.8 | 98 |
| | Spring | 19.6 | 11.0 | 8.6 | 11.8 | 16.9 | 24.3 | 34.2 | 17.1 | 1.7 | 94 |
| | Summer | 20.8 | 10.4 | 10.5 | 13.5 | 18.4 | 25.5 | 34.2 | 18.6 | 1.6 | 98 |
| | Autumn | 18.4 | 11.7 | 7.1 | 10.0 | 15.0 | 23.8 | 34.6 | 15.3 | 1.8 | 96 |
| | Workdays | 21.4 | 12.3 | 8.6 | 12.5 | 18.8 | 27.7 | 37.6 | 18.4 | 1.8 | 97 |
| | Weekends | 15.9 | 7.5 | 8.1 | 10.7 | 14.4 | 19.4 | 25.2 | 14.4 | 1.6 | 97 |
| UB | All | 11.2 | 7.1 | 4.6 | 6.4 | 9.5 | 14.0 | 19.6 | 9.5 | 1.8 | 70 |
| | Winter | 12.4 | 9.1 | 4.8 | 6.3 | 10.0 | 15.4 | 22.5 | 10.1 | 1.9 | 89 |
| | Spring | 10.4 | 6.1 | 4.6 | 6.2 | 9.0 | 12.8 | 18.3 | 9.0 | 1.7 | 100 |
| | Summer | 12.8 | 5.8 | 6.7 | 8.5 | 11.4 | 15.8 | 20.7 | 11.6 | 1.6 | 57 |
| | Autumn | 7.7 | 4.7 | 3.2 | 4.5 | 6.7 | 9.7 | 13.2 | 6.7 | 1.7 | 56 |
| | Workdays | 11.5 | 7.3 | 4.8 | 6.7 | 9.7 | 14.1 | 20.3 | 9.8 | 1.8 | 70 |
| | Weekends | 10.4 | 6.6 | 4.1 | 5.8 | 8.8 | 13.6 | 18.3 | 8.8 | 1.8 | 70 |
| DH1–3 | All | 11.7 | 8.6 | 4.2 | 6.3 | 9.7 | 14.5 | 21.1 | 9.5 | 1.9 | 94 |
| | Winter | 12.3 | 10.2 | 4.1 | 6.2 | 9.6 | 14.8 | 23.4 | 9.7 | 2.0 | 86 |
| | Spring | 12.8 | 8.2 | 5.3 | 7.4 | 10.8 | 15.9 | 23.1 | 10.7 | 1.8 | 98 |
| | Summer | 11.8 | 5.9 | 5.7 | 7.8 | 10.8 | 14.5 | 19.2 | 10.6 | 1.6 | 98 |
| | Autumn | 10.5 | 10.2 | 3.0 | 4.6 | 6.8 | 13.0 | 22.2 | 7.5 | 2.2 | 95 |
| | Workdays | 11.8 | 8.3 | 4.3 | 6.4 | 9.9 | 14.6 | 20.8 | 9.6 | 1.9 | 95 |
| | Weekends | 11.7 | 9.3 | 4.0 | 6.0 | 9.4 | 14.3 | 21.8 | 9.3 | 2.0 | 93 |
| RB | All | 7.6 | 5.4 | 2.4 | 4.0 | 6.5 | 10.2 | 14.0 | 6.1 | 2.0 | 99 |
| | Winter | 6.6 | 6.0 | 2.2 | 3.5 | 5.6 | 8.3 | 11.6 | 5.3 | 1.9 | 100 |
| | Spring | 9.1 | 6.4 | 3.5 | 5.1 | 7.4 | 11.0 | 16.6 | 7.5 | 1.9 | 99 |
| | Summer | 9.8 | 4.3 | 4.7 | 6.6 | 9.3 | 12.5 | 15.3 | 8.9 | 1.6 | 99 |
| | Autumn | 4.9 | 4.1 | 1.6 | 2.6 | 3.9 | 5.6 | 8.9 | 3.8 | 2.0 | 99 |
| | Workdays | 7.7 | 5.6 | 2.5 | 4.1 | 6.6 | 10.2 | 14.1 | 6.2 | 2.0 | 99 |
| | Weekends | 7.6 | 5.0 | 2.4 | 4.0 | 6.5 | 10.1 | 14.0 | 6.1 | 2.0 | 100 |





**Table 3.** The evaluation attributes by IAME model at the SC and the UB site, corrected to 2 significant figures.

|  | Street canyon | | | | | Urban background | | | | |
|---|---|---|---|---|---|---|---|---|---|---|
|  | $R^2$ | $MAE$ | $cRMSD$ | $r$ | $NSD$ | $R^2$ | $MAE$ | $cRMSD$ | $r$ | $NSD$ |
| All | 0.80 | 3.7 | 5.6 | 0.87 | 0.78 | 0.77 | 2.3 | 3.7 | 0.86 | 0.80 |
| Winter | 0.86 | 3.4 | 5.3 | 0.92 | 0.74 | 0.81 | 2.5 | 4.6 | 0.89 | 0.68 |
| Spring | 0.75 | 3.9 | 5.9 | 0.85 | 0.79 | 0.61 | 2.4 | 3.3 | 0.84 | 0.85 |
| Summer | 0.70 | 4.1 | 5.9 | 0.83 | 0.84 | 0.61 | 2.7 | 3.7 | 0.79 | 0.95 |
| Autumn | 0.85 | 3.4 | 5.4 | 0.9 | 0.75 | 0.85 | 1.3 | 2.0 | 0.91 | 0.83 |
| Workdays | 0.81 | 4.1 | 6.1 | 0.87 | 0.77 | 0.75 | 2.4 | 3.8 | 0.86 | 0.77 |
| Weekends | 0.72 | 3.0 | 4.3 | 0.82 | 0.82 | 0.8 | 2.1 | 3.5 | 0.85 | 0.87 |







**Table 4.** Five most successful sub-models at the SC site. The table shows only the fixed predictors with their coefficient ($\beta$, all p<0.05) and
corresponding standard error (SE). The variance inflation factor (VIF) among the fixed predictors is also shown for the 5 sub-models. The
evaluation attributes of the sub-models are shown column 6–10. The percentage of the sub-model usage and the number of data points (n)
is shown in column 11 and 12. Natural logarithm is taken for parameters with asterisk (*).

| | Fixed predictors | $\beta$ | SE | VIF | $R^2$ | MAE | cRMSD | r | NSD | % | n |
|---|---|---|---|---|---|---|---|---|---|---|---|
| 1 | *PM2.5 | 0.119 | 0.005 | 1.54 | | | | | | | |
| | *PNC | 0.313 | 0.005 | 2.89 | 0.74 | 3.7 | 5.7 | 0.87 | 0.79 | 81 | 2603 |
| | *BC | 0.223 | 0.004 | 2.17 | | | | | | | |
| 2 | *NOx | 0.236 | 0.005 | 3.79 | | | | | | | |
| | *PNC | 0.153 | 0.005 | 1.63 | 0.74 | 3.8 | 5.7 | 0.86 | 0.77 | 13 | 2629 |
| | *BC | 0.231 | 0.007 | 4.90 | | | | | | | |
| 3 | *PNC | −0.044 | 0.003 | 1.07 | | | | | | | |
| | *BC | 0.375 | 0.004 | 2.20 | 0.74 | 3.8 | 5.8 | 0.86 | 0.78 | 4 | 6622 |
| | WS | 0.201 | 0.004 | 2.15 | | | | | | | |
| 4 | *NO$_x$ | 0.250 | 0.005 | 3.09 | | | | | | | |
| | *PM$_{2.5}$ | 0.243 | 0.004 | 1.17 | 0.74 | 3.8 | 5.7 | 0.87 | 0.78 | <1 | 2596 |
| | *PNC | 0.184 | 0.005 | 3.02 | | | | | | | |
| 5 | *NOx | 0.176 | 0.005 | 3.51 | | | | | | | |
| | *PM$_{10}$ | 0.070 | 0.004 | 1.3 | 0.72 | 3.8 | 5.8 | 0.85 | 0.75 | <1 | 2713 |
| | *BC | 0.326 | 0.006 | 3.65 | | | | | | | |






**Table 5.** Five most successful sub-models at the UB site. The table shows only the fixed predictors with their coefficient ($\beta$, all $p<0.05$) and
corresponding standard error (SE). The variance inflation factor (VIF) among the fixed predictors is also shown for the 5 sub-models. The
evaluation attributes of the sub-models are shown column 6–10, corrected to 2 significant figures. The percentage of the sub-model usage
and the number of data points (n) is shown in column 11 and 12. Natural logarithm is taken for parameters with asterisk (*).

|   | Fixed predictors | $\beta$ | SE | VIF | $R^2$ | MAE | cRMSD | r | NSD | % | n |
|---|---|---|---|---|---|---|---|---|---|---|---|
|   | *CO | 0.072 | 0.027 | 1.72 |   |   |   |   |   |   |   |
| 1 | *PNC | 0.400 | 0.006 | 2.08 | 0.84 | 1.7 | 2.5 | 0.92 | 0.87 | 49 | 941 |
|   | *BC | 2.956 | 0.007 | 1.52 |   |   |   |   |   |   |   |
|   | *PNC | −0.098 | 0.005 | 1.09 |   |   |   |   |   |   |   |
| 2 | *BC | 0.398 | 0.004 | 1.44 | 0.82 | 1.9 | 2.9 | 0.91 | 0.89 | 17 | 6608 |
|   | WD-N | 0.328 | 0.006 | 1.55 |   |   |   |   |   |   |   |
|   | *NO$_2$ | 0.237 | 0.007 | 1.88 |   |   |   |   |   |   |   |
| 3 | *CO | 0.520 | 0.024 | 1.10 | 0.69 | 2.4 | 3.4 | 0.84 | 0.73 | 17 | 941 |
|   | *PNC | 0.341 | 0.010 | 2.00 |   |   |   |   |   |   |   |
|   | *CO | 0.009 | 0.000 | 1.08 |   |   |   |   |   |   |   |
| 4 | *PNC | 0.348 | 0.025 | 1.07 | 0.58 | 2.7 | 3.9 | 0.77 | 0.63 | 11 | 9757 |
|   | RH | 0.590 | 0.007 | 1.15 |   |   |   |   |   |   |   |
|   | *NO$_x$ | 0.107 | 0.006 | 2.22 |   |   |   |   |   |   |   |
| 5 | *CO | 0.182 | 0.032 | 1.72 | 0.81 | 1.9 | 3.0 | 0.90 | 0.85 | 2 | 7036 |
|   | *BC | 0.455 | 0.007 | 2.56 |   |   |   |   |   |   |   |







**Table 6.** Model evaluation comparison of deposition model derived LDSA (LDSA$_{ICRP}$), modelled LDSA by IAP (LDSA$_{IAP}$) and modelled
LDSA by IAME (LDSA$_{IAME}$) against reference measurements LDSA$_{Pegasor}$ at the SC and the UB site. Parameters with an asterisk represent
natural logarithm. The evaluation attributes of the three methods are corrected to 2 significant figures.

| | Street canyon | | | | | Urban background | | | | |
|---|---|---|---|---|---|---|---|---|---|---|
| | $R^2$ | MAE | cRMSD | r | NSD | $R^2$ | MAE | cRMSD | r | NSD |
| LDSA$_{ICRP}$ | 0.72 | 4.1 | 6.2 | 0.88 | 1.1 | 0.83 | 1.8 | 2.9 | 0.93 | 1.1 |
| LDSA$_{IAP}$ | 0.77 | 4.0 | 6.0 | 0.85 | 0.78 | 0.66 | 2.8 | 3.9 | 0.84 | 0.81 |
| LDSA$_{IAME}$ | 0.80 | 3.7 | 5.6 | 0.87 | 0.78 | 0.77 | 2.3 | 3.7 | 0.86 | 0.80 |







**Table 7.** Statistics to show temporal difference. The number of data (n), mean and standard deviation (SD) of absolute error and the
corresponding *p*-values of t-tests at the selected time windows at both sites.

| Street canyon (SC) | n | Mean | SD | t-test | *p* |
|---|---|---|---|---|---|
| Workdays | 11658 | 4.1 | 4.8 | Workdays vs Weekends | $4.13 \times 10^{-81}$ |
| Weekends | 5322 | 3.0 | 3.2 | | |
| | | | | Winter vs Spring | $3.64 \times 10^{-24}$ |
| Winter | 4023 | 3.4 | 4.2 | Winter vs Summer | $5.89 \times 10^{-5}$ |
| Spring | 2297 | 4.0 | 4.5 | Winter vs Autumn | $7.07 \times 10^{-7}$ |
| Summer | 6457 | 4.2 | 4.4 | Spring vs Summer | $6.38 \times 10^{-34}$ |
| Autumn | 4320 | 3.4 | 4.3 | Spring vs Autumn | $1.02 \times 10^{-4}$ |
| | | | | Summer vs Autumn | $2.69 \times 10^{-15}$ |
| Hour 4–10 a.m. | 4953 | 4.8 | 5.6 | Hour 4–10 a.m. vs | $2.58 \times 10^{-40}$ |
| Hour 4–10 p.m. | 4981 | 3.5 | 3.6 | 4–10 p.m. | |


| Urban background (UB) | n | Mean | SD | t-test | *p* |
|---|---|---|---|---|---|
| Workdays | 8473 | 2.3 | 2.6 | Workdays vs Weekends | $5.08 \times 10^{-8}$ |
| Weekends | 3852 | 2.1 | 2.6 | | |
| | | | | Winter vs Spring | $1.96 \times 10^{-7}$ |
| Winter | 2539 | 2.5 | 3.2 | Winter vs Summer | 0.39*** |
| Spring | 1101 | 1.9 | 3.1 | Winter vs Autumn | $1.90 \times 10^{-2}$ |
| Summer | 1628 | 2.6 | 2.4 | Spring vs Summer | $2.75 \times 10^{-9}$ |
| Autumn | 812 | 2.3 | 2.1 | Spring vs Autumn | $2.20 \times 10^{-3}$ |
| | | | | Summer vs Autumn | $1.40 \times 10^{-3}$ |
| Hour 4–10 a.m. | 3620 | 2.3 | 2.7 | Hour 4–10 a.m. vs | 0.86*** |
| Hour 4–10 p.m. | 3591 | 2.3 | 2.7 | 4–10 p.m. | |


| | n | Mean | SD | t-test | *p* |
|---|---|---|---|---|---|
| Street canyon (SC) | 11940 | 3.9 | 4.6 | SC vs UB | $8.21 \times 10^{-246}$ |
| Urban background (UB) | | 2.3 | 2.6 | (in same time period) | |

*** *p*>0.05 the null hypothesis of different distribution is rejected






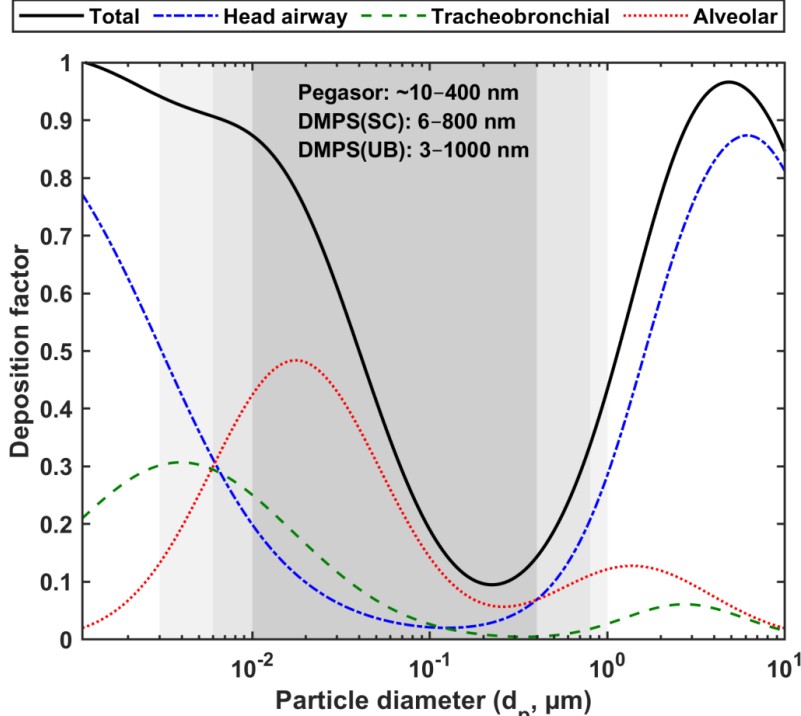

**Figure 1.** Lung deposition factor of a spectrum of particle size distribution based on the equation (ICRP, 1994). Black solid line represents the total deposition factor while blue, green and red dotted line refer to deposition factor in head airway, tracheobraonchial and alveolar region, respectively. Pegasor AQ Urban measured the alveolar LDSA concentration of particles in the ~10–400 nm size range (dark grey). DMPS at SC and UB were used to calculate alveolar LDSA in selected size fractions in the 6–800 nm and 3–1000 nm size range, respectively.




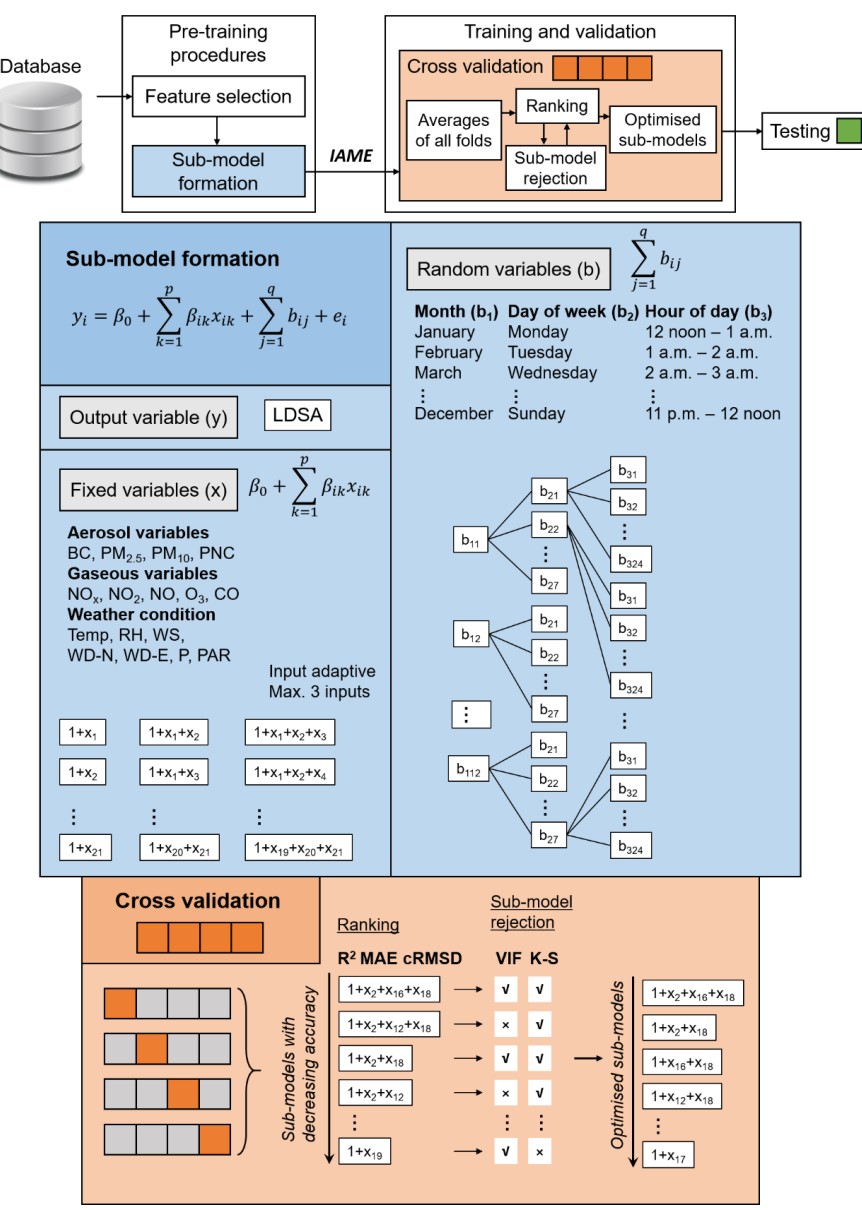

**Figure 2.** The block diagram of the proxy procedures (top). The blue and orange blocks are explanatory notes to the sections of sub-model formation and cross validation, respectively.




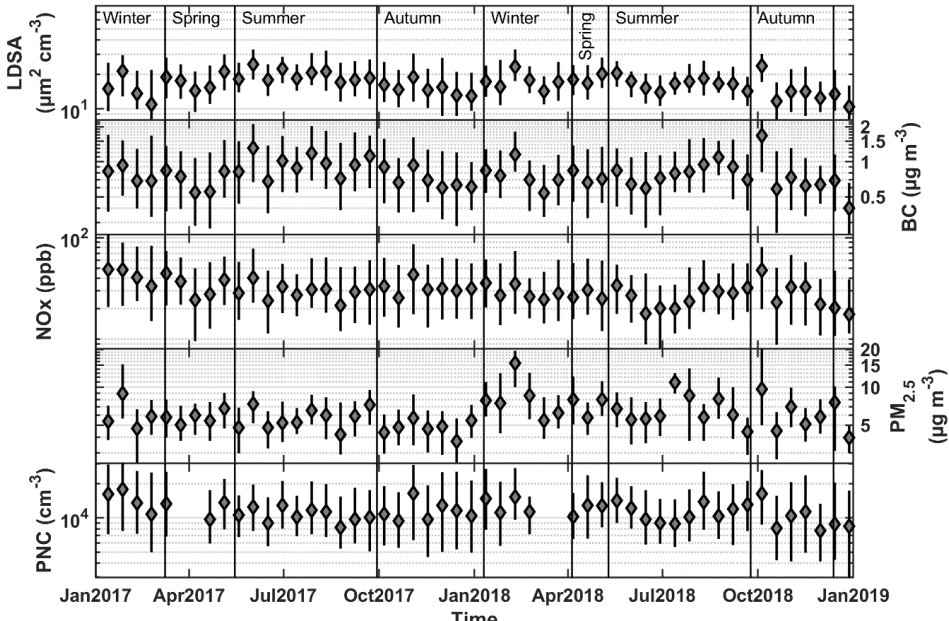

**Figure 3.** Time series of the selected air pollutant parameters (First to end row: LDSA ($\mu m^2$ $cm^{-3}$), BC ($\mu g$ $m^{-3}$), NOx (ppb), PM2.5 ($\mu g$ $m^{-3}$) and PNC ($cm^{-3}$)) at Mäkelänkatu SC site during the measurement period from 1 January 2017 and 31 December 2018. Each bar represents a period of two weeks where the shaded diamond marker is the median and the vertical error bars are the 25th and 75th percentiles. Seasons are thermally separated.








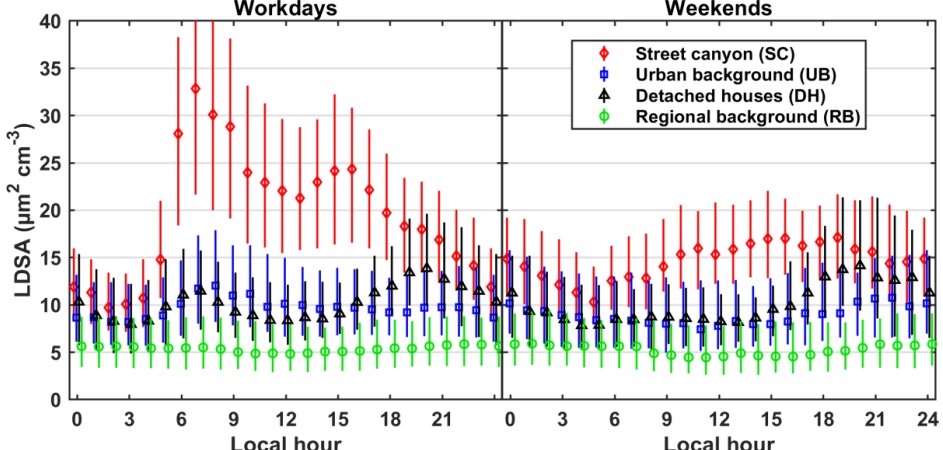

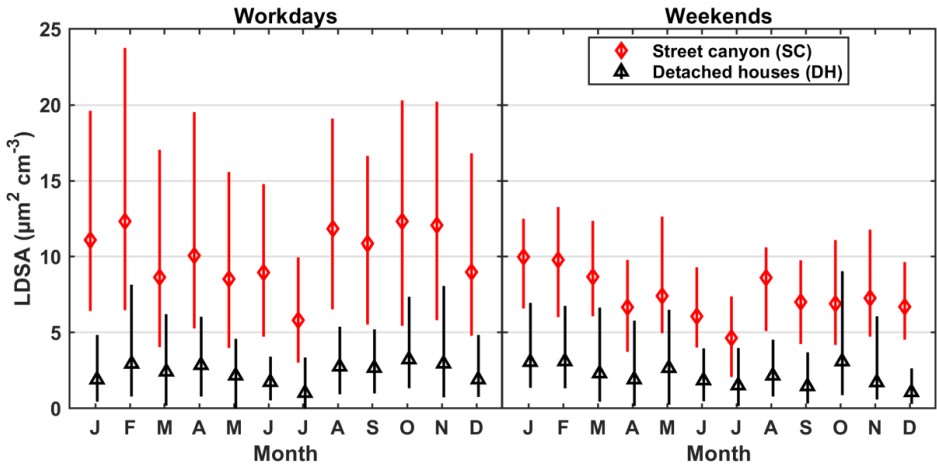

**Figure 4.** Upper panel: Diurnal cycles of LDSA concentrations ($\mu m^2\ cm^{-3}$) at SC (red diamond, 2017–2018), UB (blue square, 2017–May 2018), DH1–3 (black triangle, 2018) and RB site (green circle, 2018) on workdays and weekends with error bars of 25th and 75th percentiles. Lower panel: Monthly averages in year 2018 of local LDSA increments at the SC (red diamond) and DH1–3 (black triangle) site (LDSA concentration at the hotspot site – LDSA at RB site) on workdays and weekends with error bars of 25th and 75th percentiles.





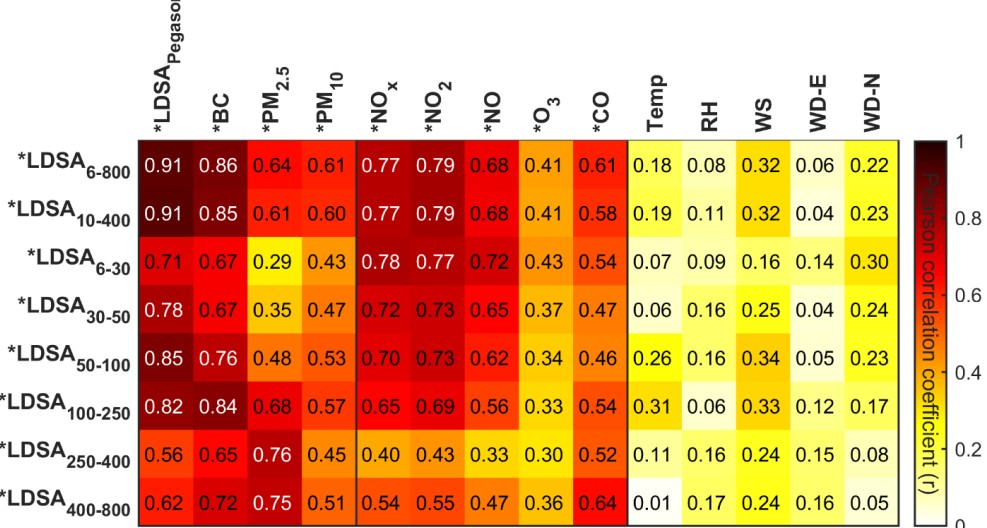

**Figure 5.** Heatmap showing Pearson correlation coefficient (r, corrected to 2 significant figures) of LDSA of different particle size sections (in nm) by ICRP lung deposition model and the other air pollutant parameters at Mäkelänkatu SC site. Dark red indicates a high correlation while pale yellow indicates a low correlation. Parameters with an asterisk represent natural logarithm. LDSA$_{Pegasor}$ represents the measured LDSA concentrations.




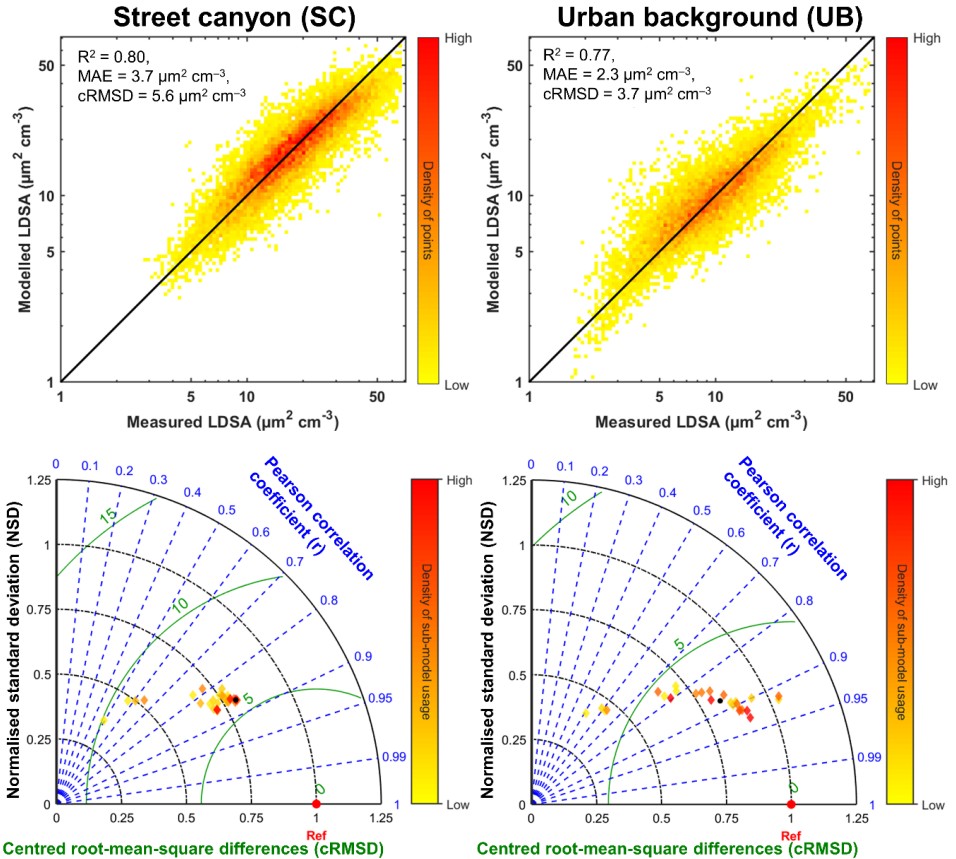

**Figure 6.** The upper panel shows the Taylor's diagrams (Taylor, 2001) at Mäkelänkatu SC site (first column) and at Kumpula UB site (second column). Each diamond marker in the Taylor's diagrams represents each sub-model used in the final estimation by IAME (solid black dot), compared with the reference data (solid red dot). Hues of colours represent how frequent the sub-model was used. The lower panel shows the scatter plots of modelled LDSA against the measured LDSA at Mäkelänkatu SC site (first column) and at Kumpula UB site (second column). Hues of colours represent the density of points on the figure.




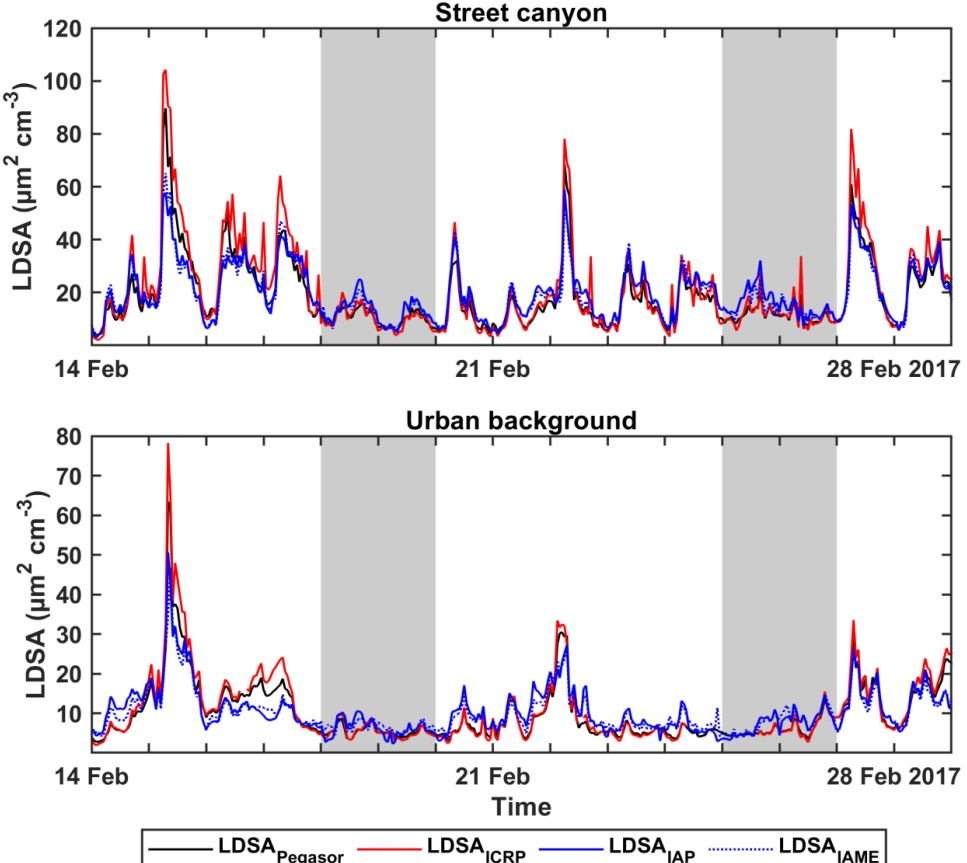

**Figure 7.** Timeseries of measured LDSA (LDSA_Pegasor, black), deposition model derived LDSA by ICRP (LDSA_ICRP, red), modelled LDSA by IAP (LDSA_IAP, blue solid line) and modelled LDSA by IAME (LDSA_IAME, blue dotted line) during a selected measurement window between 14 and 28 February 2017. Shaded regions represent weekends, otherwise workdays.







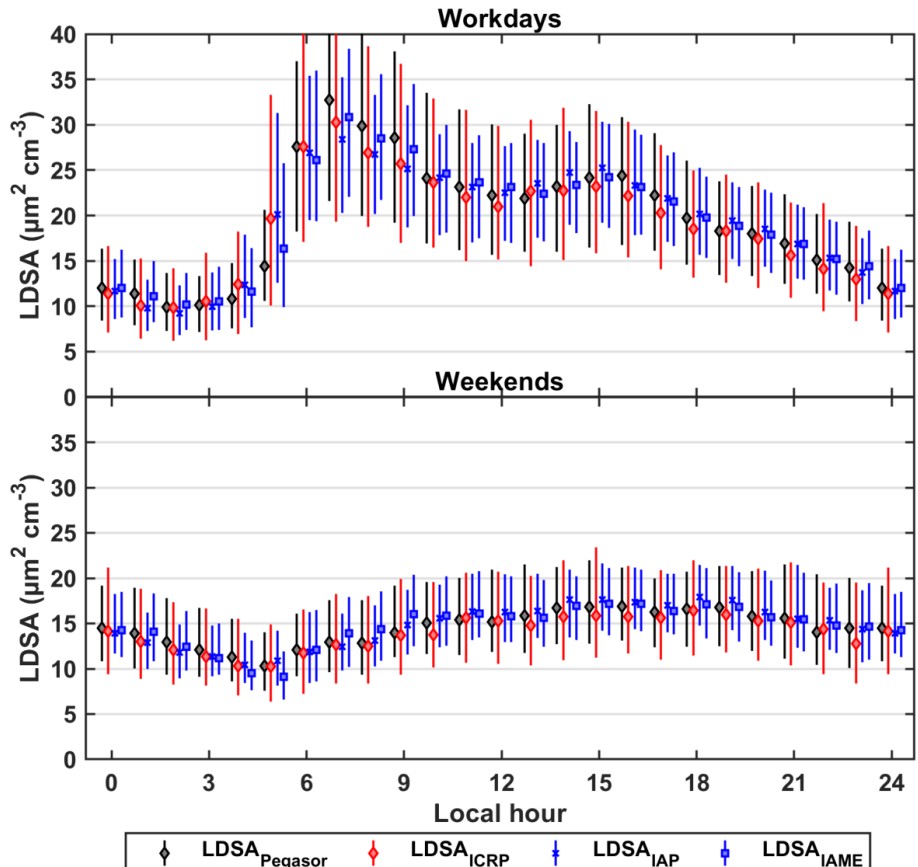

**Figure 8**. Diurnal cycles of measured (LDSA$_{Pegasor}$, black), deposition model derived (LDSA$_{ICRP}$, red) and modelled (LDSA$_{IAP}$ and LDSA$_{IAME}$, blue) LDSA concentrations with error bars of 25th and 75th percentiles on workdays (left) and weekends (right). LDSA$_{IAP}$ and LDSA$_{IAME}$ can be differentiated by their markers, cross for the former and square for the latter.

