# Peer review of "Input-adaptive linear mixed-effects model for estimating alveolar 2 Lung Deposited Surface Area (LDSA) using multipollutant datasets"

_Atmospheric Chemistry and Physics, 2021_

## Author Comment (AC2)

We show our gratitude to Anonymous Referee #1 for his constructive comments. We have revised the manuscript accordingly. Please find our point-to-point responses below.

**Response to Anonymous Referee #1's comments**

The authors observed Lung Deposited Surface Area (LDSA), which is an indicator of the adverse effects of nanoparticles on human health, in urban sites and their backgrounds, and explained their behavior and characteristics along with other parameters. As an important works of the authors, furthermore, they demonstrated to better estimate the LDSA concentrations from several widely monitored atmospheric and meteorological parameters, and characterize the statistical relationship with other parameters, by applied advanced statistical methods that combined automated input variables selection techniques with random effects. In the current that attracts public attention to the human effects of finer particles, new methods and results that better estimate this indicator, which may be better represented them than mass concentrations, from currently widely monitored parameters, have can be important implications to satisfy strong social demands in the near future.

The evaluation of the model and its usefulness are extremely and convincingly written in this manuscript. †However, I have some confusion concerns and questions regarding the interpretation of observed LDSA and some indicators analyzed by the authors. I hope the authors find my comments below useful. Therefore I would recommend the paper for publication after these clarifications.

Response: Thank you for the positive comments.

Specific comments:

Line 48-49: Clear information of particle size is misleading to the reader, because their information for particle deposition on the lungs has greater uncertainty by their various properties and also their mechanisms can be complex contribute. it necessary to clearly that these values are reference values.

Response: Thank you for your suggestion. We agree with you that the exact particle size can be inaccurate due to the reasons you mentioned. The idea of this paragraph is just to give a general information of how a pathway of a particle varies because of its particle size; therefore, we decided to remove the exact values, but to keep the terms 'coarser' and 'finer'.

Line 183-: How calibrations and corrections were made to compare the observations at several sites?

Response: The factory calibration of the instrument is based on reference number concentration aerosol, and response to LDSA is determined with 50 nm sized particles. Validation of the calibration was done by comparing the AQ Urban to other diffusion charging-based LDSA measurement devices. The instrument sensitivity with respect to LDSA is 0.215 $\mu m^2$ $cm^{-3}$ $fA^{-1}$ with the electrometer sensitivity being in the low fA range at 1 Hz operation. Better sensitivity can be achieved with longer integration times. Nominal integration time was 2 min but can be adjusted freely. According to the manufacturer, the internal precision of the AQ Urban is ± 3%, but this was not tested prior the campaign.

Table 1: Table 1: As the authors have described, previous studies of LDSA may have different target for deposition areas, which can lead to very different values. it should be clarified reviewed previous data from many sites.

Response: Thank you for the concern. Table 1 lists LDSA values from previous studies only at the alveolar region. This has been clarified in the caption of the table.

Line 330-: The contributed factors of the observed LDSA are discussed based on the time-series variations and the conclusions discussed in the previous studies observed at the same point. The characteristics of LDSA observed in this study should be more clearly based on the BC concentration data and analyzed backward trajectories, for example the effects of traffic and heating or the effects of long-distance transportation etc.

Response: Thank you for the suggestion. The observed LDSA concentrations were discussed based on Fig. 4 where we presented the diurnal cycle of LDSA. We also suggested in the original texts the possible underlying reasons for the variation, including long-range transport, human activities from traffic and heating. We improved the texts by including also the monthly frequencies of backward trajectory (Fig. S5). It shows that pollutants can be originated 600 km away from Helsinki in the winter by horizontal dispersion. Moreover, by showing the ratio LDSA/BC (Fig. S7 and S8), we also deduced that vehicular combustion emits high concentration of BC compared to its contribution to LDSA concentration.

Figure 4: Sufficient evidence is needed to explain that the value increased from the background is LDSA caused only by particles emitted in urban. In particular, it is unlikely that particles observed in background site will be observed in urban site as well without increase or decrease. Do previous studies, in particular, fully explain the long-term observations in this study?

Response: The diurnal and weekly variations in air pollutants were not clear as there were no anthropogenic activities nearby. Diurnal variation was mainly caused by the variation in the convective boundary layer height that caused mixing and dilution and not by local anthropogenic sources, which is expected for a regional background station (Luoma et al., 2021). We here made an assumption that the concentration we measured in regional background site was mostly background concentration influenced by local meteorology. The LDSA concentration at DH the after being subtracted from RB can then be regarded as the local increments to help identify the source apportionment. Similar calculation of simple subtraction was also done for example in Kuula et al. (2020) and in Jafar and Harrison (2021). To make it clear in the text, we re-organised the paragraph with one more citation.

Line 368-: The importance and implications of the ratio of LDSA to some of the parameters shown here are need to more clear.

Response: Thank you for the suggestion. The main idea of presenting the ratio of LDSA to BC, NOx, PNC and PM2.5 is to demonstrate the contribution of LDSA. For example, at the SC site, the relatively high LDSA/PM2.5 during morning peak hour reveals that the vehicular combustion emits smaller particles, which elevate the LDSA concentration but meanwhile do not substantially influence the value of PM2.5. On the other hand, the low value of LDSA/BC, LDSA/NOx and LDSA/PNC during the same peak period show the opposite. This can be explained by the fact that vehicular combustion emitted high concentration of BC, PNC and NOx compared to its contribution to LDSA concentration.

The calculation of the ratio is another way to validate the fact that urban activities, such as vehicular combustion, have different impacts on different air quality parameters. I understand that expressing the relationship in ratio is not the most straight-forward way to show the idea, but I believe it shows the contribution of LDSA from another perspective.

In particular, LDSA, which is measured by diffusion charge, is based on the relationship that the amount of charge measured is proportional to about 1.1 power of the particle size. On the other hand, PM2.5 and PINC are proportional to the 1st or 2nd power of the particle size, so in theory these three parameters are explained only by the different dimensions for diameter.

Response: Thank you for the concern. LDSA by Pegasor effectively measures the deposited surface area of ~0.01–0.4 µm and you are right that it should be proportional to about 1.1 power of the particle size after considering the deposition factor. PM2.5, however, measures mass concentration of particles smaller than 2.5 µm. PM2.5 should be proportional to the 3rd power of the particle size in the measuring range because it depends on the volume and the density of the particles. PNC in this study measures number concentration of particle sized between 0.03 to 1 µm and it should not be a function of particle size. Due to the different measuring ranges and the nature of the three parameters, I believe that these three parameters could not be explained only by the different particle sizes.

Moreover, the trend of diurnal variation in the ratio of LDSA to some components seems to be inconsistent with the relationship with the factors of LDSA explained in the previous section. This reason seems to be due to the fact that, for example, BC is a mass-based concentration, whereas LDSA is different, as mentioned earlier. Therefore, LDSA may potentially not have a linear relationship with PM2.5 and BC concentrations in theoretical, but does it affect the model constructed by these complex regressions?

Response: Thank you for raising this concern. You are right that PNC, LDSA and PM2.5 that respectively represent the number, surface and mass concentration do not have similar distribution and they do not necessarily have linear relationship. In order not to violate the assumption of the regression model, we converted the distribution of aerosol and trace gases into normal distribution by performing logarithm transformation. Also, we conducted statistical tests of the residuals of the regression model to check if they satisfy the requirements for the regression model.

Line 418-: The estimation results that showed different performance depending on the variables selected were clearly explained. Is it possible to quantitatively discuss the contribution of the parameters involved in LDSA, especially with some of the coefficients shown in Table 5?

Response: Thank you for the suggestion. Apart from counting the times the involved parameters appeared in the LDSA estimation, it is also good to refer to the Pearson correlation coefficient (r) of LDSA with those involved parameters in order to show their contribution to the LDSA estimation. Technically, the r values can effectively reflect the overall contribution. Therefore, we decided to refer some of the r values when we discuss the relative contribution in LDSA estimation (See Ln 413 and Ln 421–424). However, the coefficients shown in Table 4 and 5 could not tell the relative contribution because the ranges and magnitudes of those involved parameters are different. They, on the other hand, give the exact coefficients to calculate the output of LDSA.

Reference:

Jafar, H. A., and Harrison, R. M.: Spatial and temporal trends in carbonaceous aerosols in the United Kingdom, Atmos. Pollut. Res., 12, 295-305, https://doi.org/10.1016/j.apr.2020.09.009, 2021.

Kuula, J., Kuuluvainen, H., Niemi, J. V., Saukko, E., Portin, H., Kousa, A., Aurela, M., Rönkkö, T., and Timonen, H.: Long-term sensor measurements of lung deposited surface area of particulate matter emitted from local vehicular and residential wood combustion sources, Aerosol Sci. Tech., 54, 190-202, https://doi.org/10.1080/02786826.2019.1668909, 2020.

Luoma, K., Niemi, J. V., Aurela, M., Fung, P. L., Helin, A., Hussein, T., Kangas, L., Kousa, A., Rönkkö, T., Timonen, H., Virkkula, A., and Petäjä, T.: Spatiotemporal variation and trends in equivalent black carbon in the Helsinki metropolitan area in Finland, Atmos. Chem. Phys., 21, 1173-1189, https://doi.org/10.5194/acp-21-1173-2021, 2021.

---

## Author Comment (AC3)

We show our gratitude to Anonymous Referee #2 for his constructive comments. We have revised the manuscript accordingly. Please find our point-to-point responses below.

**Response to Anonymous Referee #2's comments**

This research looked how to model missing lung-deposited surface area data from both street canyons and urban background environments. This work showed that more research is needed in this area to better predict these gaps in data, but provides correlations between their revised model and real-world data.

Response: Thank you for the good summary of the manuscript.

Comments to the Authors

Line 45 – in particular to respiratory systems. "System" needs an "s"

Response: It has been revised accordingly.

Is there a reference for the particle deposition assumptions from line 48 – 49 of 5-30 um particles?

Response: Thank you for the question. In response with a suggestion by another referee, we decided to remove the exact values of the particle size, but to keep the terms 'coarser' and 'finer' to avoid misleading. The reference for these two sentences is Gupta and Xie (2018).

Lines 50 – 58 – the discussion of COVID in the introduction does seem directly relevant to the study, at least in the context discussed here. It could be said that the surface area of the particles could act as transport vectors for viruses and bacteria, and therefore, the commonly monitored particle matter is number concentration and mass concentration, …" picking up on line 59.

Response: Thank you for the suggestion. In order not to divert the focus of the article, I agree that we should remove the discussion of COVID in the introduction. The text is updated according to your suggestion.

Methods section: What are the instruments' allowed variance/uncertainty (+/- 5%, 2%?)

Response: According to the manufacturer, the internal precision of the AQ Urban is ± 3%, but this was not tested prior the campaign. This sentence is now inserted in the method section.

Is there a reference for quantifying LDSA from derivations of particle size distribution?

Response: A reference by Hinds (1999) has been inserted for quantifying LDSA from particle size distribution and lung deposition curve model.

The introduction currently focuses on what LDSA is, how they move through the respiratory system, how they are currently measured, and other models that have tried to do this similar modelling. Although the introduction is already quite lengthy, it does not explain why the gaps in data are so critical to understand. Line 106/107 only mentions that these instruments sometimes "lose" data and it should be accounted for, but the authors need to address why the data needs to be accounted

for. This can then be used as a central talking point in the conclusion as to how this model is helpful to the community.

Response: Thank you for the suggestion. It is very important to address why the data needs to be accounted for. The paragraph is improved by explaining that another aim of the paper is to use the statistical models as virtual sensors after they are validated. They are useful when the actual instruments are under long-term maintenance. Due to the health effects LDSA has demonstrated, it is useful for researchers to have continuous measurements for research purposes (See Ln 98-102).

To help concentrate the introduction, the discussion of the different types of LDSA measurement techniques (Lines 82 -103) could be summarised in two or three sentences.

Response: The paragraph is now trimmed off into a few sentences (See Ln 77-88).

Line 108 - Possibly changing the end of that sentence to "… under certain circumstances, such as traffic activities."

Response: We changed the wording '…under certain circumstances' to '…in traffic environments.' to avoid misunderstanding.

Are there other circumstances that correlate well? Are there areas that do not correlate well that data correction cannot be used for?

Response: Based on the paper we cited, the correlation of LDSA with BC and NOx is high ($R^2>0.7$) in street canyon, but not as high in urban background and detached housing. Therefore, we changed the wording '…under certain circumstances' to '…in traffic environments' to avoid misunderstanding. In my understanding, the correlation of LDSA with BC and NOx is not a factor of data correction.

Line 205 – The LDSA study that showed large accumulation mode particles should be similar to the street canyon area of the study. What does this mean for this study?

Response: Accumulation mode particles refer to particles sized 500nm to 2000nm. This echoes to the beginning of the sentence that we aimed to stress that the impact of larger particles (>400 nm) to alveolar LDSA could be significant but the Pegasor AQ Urban can only measures LDSA up to 400nm.

Lines 203- 209, due to these findings, are there further limitations on this work? What does the India study contribute to this work? What fraction of the particles measured are assumed to be above 400 nm at these locations? Presumably, the street canyon site would have more near traffic particles above 400 nm, whereas the urban background would be more influenced by long range transport particles. How can you discern these artifacts measured in this campaign?

Response: Thank you for raising the concern. It is rather difficult to estimate the fraction of particles >400nm. It depends on whether $PM_{2.5}$ episode is ongoing or when there are many particles from very low-quality residential burning in detached housing areas. In response to the artefacts, we also calculated LDSA by deposition curve ($LDSA_{ICRP}$) and compared it with the measured LDSA ($LDSA_{Pegasor}$, see Fig. 5). $LDSA_{Pegasor}$ correlates fairly with $LDSA_{400-800}$ (r=0.62) but very well with $LDSA_{6-800}$ (r=0.91). It indicates that although LDSA measured by Pegasor could not detect particles > 400nm, it did not have significant effects on this whole dataset. We included the India study because the study

compared the LDSA concentration in India and Helsinki. Also, it showed that larger particles might have a significant impact on LDSA concentration. We improved the paragraph by reorganising the text in Ln 196-201. We added one more limitation on this work.

Line 206 – "environment" needs an "s" to make it "environments".

Response: Thank you for the comment. It has been revised accordingly.

Line 231 – is there any indication of what caused the outliers, or how many were deleted from the dataset?

Response: Thank you for the question. The outliers were detected by commonly used interquartile range (IQR) method. The underlying reasons can be potential measurement errors due to extreme weathers or instrument defects. Overall, there were 0.73% and 0.99% data points identified as outliers in street canyon SC and urban background UB site, respectively. This information is now updated in the manuscript (See Ln 223).

Line 383-391 – can the correlations be quantified here ( i.e. $R^2$ = …), for at least a few of them?

Response: Thank you for the comment. We agree that it is necessary to include some of the correlations in the paragraph. It has now been done.

Figure 6 – Could the authors give a 1-2 sentence (further) explanation of the Taylor diagram. It is an interesting way to summarize statistical correlations. A simple solution would be to put in brackets the color of the lines that the correlate to within the text so:

Line 407 would become – "The five mostly used sub-models are shown in Figure 6 where r (Blue contours) is 0.85–0.87, ð•'•ð•' ..ð•' €ð•' †ð•• (Green contours) is 5.67–5.77 µm2 cm–3 407 and ð•'•ð•' †ð•• (Black axis) is 0.75–0.79, and also shown in Table 4"

Response: Thank you for the suggestion. The sentence is now revised accordingly.

Also, the figure captions has the two panels are reversed in order (scatter plots are top panel and the Taylor diagrams are the lower)

Response: We apologize for this mistake. The caption is now fixed.

Line 437 – Is there a way to correct for the over/underestimation for sharp peaks? How important are these peaks for contributing to the motive of the study? If these peaks are used to determine 1 hour- exposure levels, they would need to be fairly accurate, but if they are to close the gap for monthly averages, the inaccuracy is less important.

Response: To improve the accuracy of the model, I would suggest to include more input variables. One could be the number of road traffic because most of the over/underestimation took place in the morning peak hours. However, the information of traffic is not commonly available next to the measurement station. Therefore, it would sound a bit impractical.

One of the aims of the model is to utilize the model as virtual sensors. The results of the individual data points might not be that promising; however, if we use the model as virtual sensor in a long term, the overall accuracy is fairly high (SC: $R^2$=0.80; UB: $R^2$=0.77).

Is the amount of work needed to model the missing data worth the inaccuracy of it? If the model is over or underestimating by up to 100%, what is the contribution of this modelled data to anyone using the real-world data? This goes back to the rational behind the project and its contribution to the scientific community

Response: Apart from filling up missing data, another important aim is to use the model as virtual sensors. The results of the individual data points might not be that promising; however, if we use the model as virtual sensor in a long term, the overall accuracy is fairly high (SC: R2=0.80; UB: R2=0.77). The underestimation of almost 100% is only for one data point. Our intention is to show some of the limitations of the model, but not to focus on just one individual data point. In order to get the focus back to the aim, we decide to remove that misleading sentence (See Ln. 432-441).

Line 504 - In the street canyon scenario, IAME is less likely to accurately model instantaneous peaks, meaning that using this for determining the least polluted route to take in an urban area might not be the best application for this model, as it would not reliably be showing what is happening in real time.

Response: Thank you for the suggestion. IMAE could capture the diurnal pattern and most of the peaks. You are also right that it could over/underestimate some of the individual sharp peaks. Every model has uncertainties. The intention to deploy this model to GreenPaths is that currently GreenPaths do not take LDSA in account. Therefore, it would at least give a fairly good estimate by using IMAE proxy, not to mention the good long term accuracy. But for sure, we would love to investigate more to improve the model, for example making use of land use regression to give spatio-temporal estimation in the future. To avoid confusion, we remove this example as one of the possible applications of IMAE.

Reference:

Gupta, R., and Xie, H.: Nanoparticles in daily life: applications, toxicity and regulations, J. Environ. Pathol. Tox., 37, https://doi.org/10.1615/JEnvironPatholToxicolOncol.2018026009, 2018.

Hinds, W. C.: Aerosol technology: properties, behavior, and measurement of airborne particles, John Wiley & Sons, 1999.

---

## Author Response (AR2)

We show our gratitude to Anonymous Referee #1 for his constructive comments. We have revised the manuscript accordingly. Please find our point-to-point responses below.

**Response to Anonymous Referee #1's comments (round 2)**

The authors' manuscript has been considered according to the suggestions of the reviewers and has appropriately revised. But there is still one issue which is not well addressed. Therefore I would recommend the paper for publication after these revise.

The authors replied that the relationship with LDSA concentration for some parameters is not always linear. I agree with that. Therefore, while the requirements for the model may be met, I am concerned that the difference in the time series variation of each ratio may not be able to describe the difference in potential contribution to the particles. In particular, the authors describe that ultrafine particles emitted from vehicles such as BC increase the LDSA concentration, while rush hour contribute to those BC concentrations rather than LDSA concentration from differences in temporal variations. This can be quite confusing. Certainly, the mathematical contribution for each value may be explain to more contributed by the BC concentration, but is it due to the different properties of each parameter? Or is it explained by the time-dependent increase in particle size of the ejected particles during the rush hour?

Response: Thank you for the comment. The reviewer is right that the difference in time series variation may not be able to fully describe the difference in potential contribution to the particles. During rush hour, the emitted particles from traffic are usually smaller in size, dominated by diameter less than 50 nm. In this size range, the lung deposition factor that constitutes LDSA can be up to 0.5 (Figure 1). As time goes by, the particles may grow to a larger size that has another deposition factor. Therefore, the ratio between LDSA and different parameters are dependent on both the different properties of each parameter (e.g. the lung deposition factor), and the time-dependent increase in particle size (e.g. new particle formation), as the reviewer pointed out. However, based on the data/results in this manuscript, we could not draw a concrete conclusion of the underlying reasons for the difference in the time series variation of each ratio. The original text can only partly show the reasons. To avoid confusion, we inserted a few sentences in Section 4.2 to describe more on what we expected to tell by showing the ratios of different parameters.